# A Tourist-Based Framework for Developing Digital Marketing for Small and Medium-Sized Enterprises in the Tourism Sector in Saudi Arabia

Rishaa Abdulaziz Alnajim * and Bahjat Fakieh

Department of Information Systems, Faculty of Computing and Information Technology,
King Abdulaziz University, Jeddah 21589, Saudi Arabia; bfakieh@kau.edu.sa
*   Correspondence: rshanjm@gmail.com

**Abstract:** Social media has become an essential tool for travel planning, with tourists increasingly using it to research destinations, book accommodation, and make travel arrangements. However, little is known about how tourists use social media for travel planning and what factors influence their intentions to use social media for this purpose. This thesis aims to understand tourists' intentions to use social media for travel planning. Specifically, it investigates the factors influencing tourists' intentions to use social media for planning travel to Saudi Arabia. It develops a machine learning (ML) classification model to assist Saudi tourism SMEs in creating effective digital marketing strategies for social media platforms. A survey was conducted with 573 tourists interested in visiting Saudi Arabia, using the Design Science Research (DSR) approach. The findings support the tourist-based theoretical framework, showing that perceived usefulness (PU), perceived ease of use (PEOU), satisfaction (SAT), marketing-generated content (MGC), and user-generated content (UGC) significantly impact tourists' intentions to use social media for travel planning. Tourists' characteristics and visit characteristics influenced their intentions to use MGC but not UGC. The tourist-based ML classification model, developed using the LinearSVC algorithm, achieved an accuracy of 99% when evaluated using the K-Fold Cross-Validation (KF-CV) technique. The findings of this study have several implications for Saudi tourism SMEs. First, the results suggest that SMEs should focus on developing social media content that is perceived as useful, easy to use, and satisfying. Second, the findings suggest that SMEs should focus on using MGC in their social media marketing campaigns. Third, the results suggest that SMEs should tailor their social media marketing campaigns to the characteristics of their target tourists. This study contributes to the literature on tourism marketing and social media by providing a better understanding of how tourists use social media for travel planning. Saudi tourism SMEs can use the findings of this study to develop more effective digital marketing strategies for social media platforms.

**Keywords:** digital marketing; machine learning; Saudi tourism; SMEs; travel planning

## 1. Introduction

Tourism is a primary industry that generates significant economic and social benefits for many countries worldwide. The World Travel and Tourism Council (WTTC) declared that the travel and tourism sector contributed 7.6% to the global Gross Domestic Product (GDP) in 2022, an increase of 22% from 2021 [1]. Tourism positively affects culture, healthcare, education, communications, infrastructure, and transportation [2]. Tourism generates economic growth and long-term sustainability by providing jobs and directing exports.

In 2016, Saudi Arabia launched Vision 2030 to develop diverse economies and focus on expanding the non-oil sectors and generating long-term sustainability. Vision 2030 aims to develop numerous sectors, including tourism [3]. Until recently, tourism in Saudi Arabia was limited to Muslims visiting the holy towns of Makkah and Madinah for Umrah and Hajj [4]. Today, Saudi Arabia is issuing tourist visas for tourists worldwide, transforming

Saudi Arabia from a religious tourism destination for two to three months to a year-round tourism destination [5]. According to the Saudi Tourism Ministry's tourism indicators report for the third quarter of 2022, the number of tourists exceeded 22 million, with total spending of SAR 26 billion [5]. The Saudi Tourism Ministry works with other Saudi National Government Organizations to develop the tourism sector, which includes attracting heavy investments and arranging global events and festivals [5].

Moreover, Saudi Arabia empowers small and medium-sized enterprises (SMEs) as a crucial pillar in economic growth, development, and job opportunities [6]. Saudi Arabia aims to raise SMEs' contribution to GDP to 35% by 2030 [3]. SMEs are an essential part of the tourism industry, as they often provide personalized and unique experiences that larger businesses may be unable to offer [7]. SMEs include hotels, restaurants, tour operators, travel agencies, and other businesses that provide tourist services. However, SMEs in the tourism sector face several challenges that can make it difficult for SMEs to compete in the market and achieve long-term sustainability [8]. One of the main challenges is the rapidly changing digital landscape, which requires SMEs to constantly update their digital marketing strategies to keep up with new technologies and platforms [9]. Another challenge is the increasing competition for online visibility, as larger businesses with more significant resources and marketing budgets can dominate search engine rankings and social media platforms [9].

Additionally, SMEs may need help to create high-quality digital content and engage effectively with customers online, particularly if they lack the necessary skills or resources. Finally, SMEs may need help measuring the effectiveness of their digital marketing campaigns, which can make it difficult to justify continued investment in this area. Despite these challenges, digital marketing presents significant opportunities for tourism SMEs to reach new audiences, build brand awareness, and drive bookings and revenue [10].

Recently, social media platforms have become a popular tool for travel planning. Many tourists rely on social media content and recommendations to decide where to go, what to eat, and where to stay. According to a recent survey by Statista, 75% of tourists use social media to research and plan their trips. In comparison, 47% of tourists depend on the experiences of friends and family [11]. Tourists use social media primarily to seek destination information, with visual content such as photos and videos being particularly influential. Social media platforms such as Facebook, Instagram, and Twitter are the most popular sources of destination information for tourists, who also use social media to book hotels and flights and purchase travel-related products and services. Moreover, social media is vital for sharing travel experiences and connecting with other travelers, and significantly impacts tourists' satisfaction and destination loyalty [12].

The principal objective of this study is to assist Saudi tourism SMEs in developing effective digital marketing strategies for social media platforms by proposing a tourist-based theoretical framework for identifying the factors that influence tourists' intentions to use social media platforms for planning travel to Saudi Arabia. Also, it demonstrates the relationship between these factors. Moreover, this study explains the influence of social media content on tourists' intent to use social media platforms for travel planning by developing the tourist-based ML classification model, which segments tourists based on whether they prefer MGC or UGC when using social media for planning travel to Saudi Arabia. The results of this study could assist and benefit the Saudi tourism SMEs in developing an effective content marketing strategy on social media by utilizing ML. The study objectives can be summarized as follows:

- To propose a tourist-based model for identifying tourists' intentions to use social media for planning travel to Saudi Arabia;
- To assist Saudi tourism SMEs in developing an effective digital marketing strategy on social media by developing the tourist-Based mL classification model, which segments tourists based on whether they prefer MGC or UGC when using social media for planning travel to Saudi Arabia.

Using ML to analyze and understand the intention of tourists to use social media for planning travel to Saudi Arabia is an important area of research that might assist in identifying many factors that influence the Saudi tourism industry. The following research questions are aimed at exploring the tourists' intentions to use social media for planning travel to Saudi Arabia and assisting Saudi tourism SMEs in developing an effective digital marketing strategy on social media by utilizing ML:

- What variables impact tourists' intentions to use social media platforms for planning travel to Saudi Arabia?
- What is the impact of the tourists' characteristics on their intentions to use MGC and UGC on social media platforms for planning travel to Saudi Arabia?
- What is the impact of visit characteristics on tourists' intentions to use MGC and UGC on social media platforms for planning travel to Saudi Arabia?
- How can ML be used to improve the digital marketing strategies of Saudi tourism SMEs on social media platforms?
- What recommendations can help to improve the digital marketing strategies of Saudi tourism SMEs on social media platforms?

Answering these research questions can provide valuable insights into the factors influencing tourists' intentions to use social media for planning travel to Saudi Arabia. Such insights can help Saudi tourism SMEs utilize ML to develop effective social media marketing strategies that are effective and reflect the needs and tastes of their target audience.

This study is structured as follows: the literature review, research methodology, data analysis and results, and discussion are presented in. Finally, the conclusion will be placed in Section 6.

## 2. Literature Review

Few studies have been conducted on digital marketing in the Saudi tourism sector, particularly within the context of tourism SMEs. However, several studies have discussed social media as a digital marketing channel in business, SMEs, and tourism marketing in various countries worldwide. The following section highlights some of the studies in the field.

### 2.1. Digital Marketing

Several investigations have been conducted to evaluate the increasing importance of digital marketing and its evolving strategies based on consumer preferences. Social media and websites are increasingly popular in marketing operations since they provide more personalized connections between consumers and vendors, strengthening the relationship and enhancing market involvement [13]. Online business marketing professionals create a content network on popular digital platforms like Facebook, Twitter, and Instagram, then switch to business-to-business social media platforms like LinkedIn to publish more specialized content [14].

### 2.2. Social Media as a Channel of Digital Marketing

Social media marketing has branched out from social media as a new marketing strategy that relies on electronic word of mouth (eWOM), promoting services and products via social media to reach many targeted consumers [15]. A social media marketing strategy is capable of reaching more consumers than conventional marketing strategies due to its flexibility in reaching consumers more cheaply than traditional media such as TV, radio, and newspapers [16].

The findings of previous studies have shown a relationship between business performance and social media usage. Utilizing social media for business has been proven to have the potential effects of reducing costs, reaching target consumers, increasing sales, and improving consumer service [17]. Social media marketing activities implicitly impact consumers' satisfaction through social identity and perceived value. Simultaneously, social

identity and perceived value explicitly influence consumers' satisfaction, affecting the desire to stay, the intention to join, and the intention to buy [18].

Social media data can be an essential source of customer analysis, market research, and crowdsourcing new ideas. Capturing and creating value through social media data represents the development of a new strategic resource to improve marketing outcomes [19]. It is essential to consider social media's future in consumer behavior and marketing, since it has become a strong marketing and contact channel for businesses, organizations, and institutions. Consequently, businesses need to switch from traditional to digital marketing to remain competitive in today's market [20]. Social media and digital marketing are becoming strategic tools for building brand awareness and marketing campaigns. Additionally, consumers now rely significantly on social media platforms as a source of information when making purchasing decisions [21]. According to a study by GlobalWebIndex (2018), 54% of consumers use social media platforms to research products before purchasing, highlighting the importance for businesses of having a solid social media presence [22].

Overall, social media has become an important channel for digital marketing. It allows businesses to contact their target audience more effectively while giving data on essential consumer habits and preferences. Furthermore, social media can help companies to build relationships with their customers by providing personalized content and engaging with them regularly, increasing customer loyalty and retention rates.

*2.3. Social Media Marketing in SMEs*

SMEs' most common use of social media is to promote their products or services, but they can also use it to build customer relationships, provide customer service, and gather feedback. However, many SMEs do not understand how to utilize social media to achieve their business objectives, which indicates a need for more education and training on social media marketing, since using social media effectively can provide a competitive advantage in the marketplace [23]. For example, SMEs need to track the results of their social media campaigns to target their marketing messages to specific audiences.

On the other hand, SMEs face several potential barriers and challenges in using social media marketing, such as experimentation with different strategies, the lack of evidence concerning best practices, launching campaigns without a clear goal, consumers' public assessment of the quality of services, and the quality of social media interactions being critical factors [24].

Furthermore, SMEs face the challenge of creating a consistent and engaging social media presence, which requires time, effort, and creativity [25]. SMEs may need to hire specialists to manage their social media accounts or outsource their marketing to a third-party agency [26]. Another challenge for SMEs is measuring the return on their social media marketing investment. While tracking metrics such as comments, likes, and shares is relatively easy, it can be more challenging to determine how these metrics translate into sales or revenue [26]. SMEs may need to utilize tools like Facebook Insights or Google Analytics to measure the effect of social media marketing on investment returns [27].

These challenges may make it challenging for SMEs to use social media marketing successfully, but those that do so will have a significant competitive advantage. Numerous experiments have investigated SMEs' challenges in social media marketing to provide effective solutions that might confront the challenges and develop marketing strategies. In 2020, a sample of 35 SMEs from various industries in Trinidad and Tobago, including clothing, food and beverages, and sporting goods, participated in a study on using social media platforms as a marketing tool [28]. The interviews were conducted to understand the SMEs' strategies in social media marketing and how their experiences impacted their overall business performance. As a result, a structured framework was proposed for SMEs to utilize social media effectively in their marketing efforts. This framework included monitoring social media output, carefully evaluating social media platforms before selecting one, and developing a comprehensive marketing strategy.

*2.4. Social Media Marketing in Tourism*

Tourism SMEs can also improve the quality of their services and design new services for tourists based on social media data. Tourists use social media for travel planning, booking accommodation, reservations, confirming and cancelling, inquiring about itineraries and packages, reading other tourists' reviews, and sharing their travel experiences with others by rating, writing reviews, commenting, and sharing photos [29]. Several studies have examined the content generated by tourists, often known as UGC, on social media and its use to enhance tourist marketing. It was discovered that social media advertising strategies such as influencer marketing and UGC might disseminate sustainable tourism [28].

However, there are also potential barriers and challenges to utilizing social media for marketing tourism. For example, social media platforms are continuously developing, and tourism SMEs must keep up with the newest trends and features to remain relevant [30].

Furthermore, social media is time-consuming and needs a significant expenditure of resources. Tourism SMEs may use several tactics to efficiently handle these problems, such as building a social media calendar to prepare content ahead of time, employing analytics tools to measure performance indicators, and engaging with influencers to reach a larger audience. Tourism SMEs can also keep up with the newest trends and features on social media platforms by attending conferences and seminars or employing social media professionals [31].

## 3. Tourist-Based Theoretical Framework

Tourism marketers have tried to keep up with new technology development and found positive and beneficial results [14]. Increasing the efficiency of utilizing social media in tourism marketing would bring several benefits to businesses [14]. Consumer behavior on social media has been studied in several research studies. The present study investigates the behavior of tourists on social media by identifying their utilization of social media as a new technology when planning to travel to Saudi Arabia and determining the factors that influence their decisions.

The technology acceptance model (TAM) is employed in this study, which is designed to describe the perception of accepting new information systems and is rooted in two psychological theories: the theory of reasoned action (TORA) and the Theory of Planned Behavior (TOPB) [32]. The proposed tourist-based theoretical framework extends the TAM by including more predictions to address the identified research gap. The tourist-based theoretical framework examines the influence of perceived usefulness (PU), perceived ease of use (PEOU), satisfaction (SAT), marketing-generated content (MGC), and UGC on tourists' intentions to use social media when planning travel to Saudi Arabia. The tourists' socioeconomic characteristics and visit characteristics were added as moderated variables. The proposed tourist-based theoretical framework is shown in Figure 1.

In the context of travel planning, PEOU and PU significantly positively affect tourists' intentions to use social media to plan travel [33]. PEOU refers to "the degree to which tourists perceive that using social media for planning travel to Saudi Arabia is simple, easy, and effortless" [33]. PU refers to "the degree to which tourists perceive that using social media for planning travel to Saudi Arabia might improve their travel plans and provide them with more best travel options" [33]. SAT is tourists' satisfaction with the tourism content about Saudi Arabia on social media platforms. SAT positively influences tourists' intentions to use social media platforms for planning travel to Saudi Arabia [34]. MGC is the content generated and shared by marketers in Saudi tourism SMEs on social media platforms to engage with consumers or to offer and promote products, services, and achievements [35]. UGC is the content that internet users have created and shared on social media platforms regarding travelling to Saudi Arabia [35]. UGC comes in different formats, such as texts, discussions, blogs, images, audio, videos, consumer reviews, and any other ways that allow users to generate content and engage with others. MGC and UGC positively influence tourists' intentions to use social media platforms for planning travel to Saudi Arabia [36]. As a consequence, the study proposes the following hypotheses:

**Hypothesis 1 (H1).** *PU positively affects tourists' intentions to use social media for planning travel to Saudi Arabia.*

**Hypothesis 2 (H2).** *PEOU positively affects tourists' intentions to use social media for planning travel to Saudi Arabia.*

**Hypothesis 3 (H3).** *SAT positively affects tourists' intentions to use social media for planning travel to Saudi Arabia.*

**Hypothesis 4 (H4).** *MGC positively affects tourists' intentions to use social media for planning travel to Saudi Arabia.*

**Hypothesis 5 (H5).** *UGC positively affects tourists' intentions to use social media for planning travel to Saudi Arabia.*

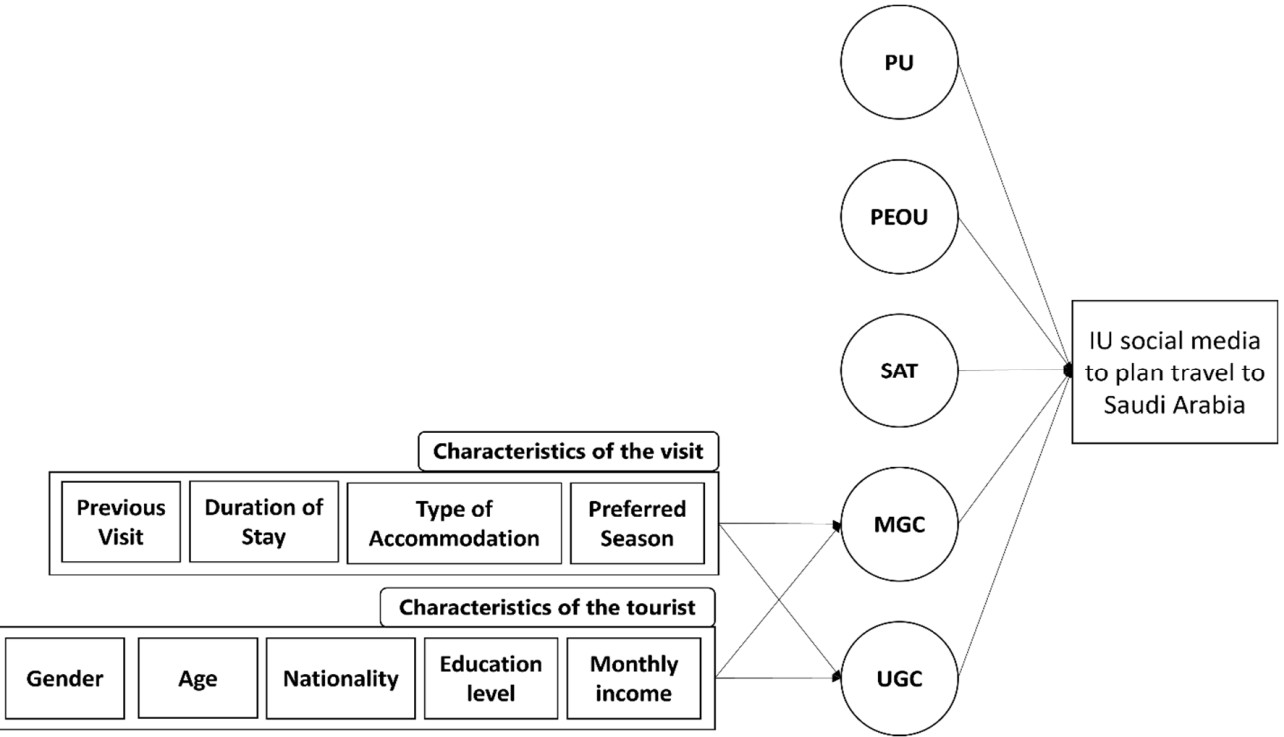

**Figure 1.** Tourist-based theoretical framework (source: created by the authors).

Tourists' socioeconomic characteristics, such as their gender, age, nationality, education level, and monthly income, could be significant predictors of the influence of MGC and UGC on tourists' intentions to use social media for planning travel to Saudi Arabia. Furthermore, the correlation between these variables and tourists' motivations has been discovered in previous studies. Including these predictors as control variables in hypothesis testing enhances its reliability regarding tourists' socioeconomic characteristics [37].

Tourists' intentions to use MGC and UGC on social media platforms for planning travel to Saudi Arabia could differ by gender. Also, the differences across tourist generations influence their intention to use MGC and UGC on social media platforms for planning travel to Saudi Arabia, with younger generations usually more aware and ready to use it than their elders [23]. As a result, this study proposes the following:

**Hypothesis 6 (H6).** *The effect of MGC on tourists' intentions to use social media for planning travel to Saudi Arabia will differ by gender.*

**Hypothesis 7 (H7).** *The effect of UGC on tourists' intentions to use social media for planning travel to Saudi Arabia will differ by gender.*

**Hypothesis 8 (H8).** *The effect of MGC on tourists' intentions to use social media for planning travel to Saudi Arabia will differ by age.*

**Hypothesis 9 (H9).** *The effect of UGC on tourists' intentions to use social media for planning travel to Saudi Arabia will differ by age.*

Previous studies have shown that the intentions and behaviors of social media users differ depending on their nationality [38]. Hence, tourists' intentions to use MGC and UGC on social media platforms for planning travel to Saudi Arabia might be influenced by their nationalities. For instance, tourists from some countries might be more likely to rely on MGC than UGC on social media for planning travel to Saudi Arabia [39]. Therefore, tourism marketers in Saudi tourism SMEs need to understand the cultural differences among tourists from different countries when developing their marketing strategies. As a result, this study proposes the following:

**Hypothesis 10 (H10).** *The effect of MGC on tourists' intentions to use social media for planning travel to Saudi Arabia will differ by nationality.*

**Hypothesis 11 (H11).** *The effect of UGC on tourists' intentions to use social media for planning travel to Saudi Arabia will differ by nationality.*

The educational level of tourists may influence their intentions to use MGC and UGC on social media platforms when planning travel to Saudi Arabia; previous studies have shown that tourists with higher levels of education tend to use social media platforms more frequently, as they are more familiar with social media and trust the information available [40]. In addition, the monthly income of tourists also plays a role in their travel planning. Tourists with median incomes are more likely to have stronger travel intentions, as they can afford to travel to Saudi Arabia [40]. Therefore, this study suggests the following:

**Hypothesis 12 (H12).** *The effect of MGC on tourists' intentions to use social media for planning travel to Saudi Arabia will differ by education level.*

**Hypothesis 13 (H13).** *The effect of UGC on tourists' intentions to use social media for planning travel to Saudi Arabia will differ by education level.*

**Hypothesis 14 (H14).** *The effect of MGC on tourists' intentions to use social media for planning travel to Saudi Arabia will differ by monthly income.*

**Hypothesis 15 (H15).** *The effect of UGC on tourists' intentions to use social media for planning travel to Saudi Arabia will differ by monthly income.*

The characteristics of the visit, such as previous visits, the season, length of stay, and type of accommodation, could be significant predictors of the influence of tourists' intentions to use MGC and UGC on social media for planning travel to Saudi Arabia. A tourist's previous experience is also an essential variable that influences tourists' intentions and behavior [41]. Tourists who have previously visited Saudi Arabia are more likely to use MGC and UGC on social media for travel planning than those who have not because they are more familiar with the destination and seek information about specific places and activities. Also, tourists who prefer to travel in the summer may rely on UGC more than MGC because they are looking for information about other tourists' experiences in the summer. The season in which a tourist travels can also influence their use of MGC and UGC on social media [42]. Therefore, this study suggests the following:

**Hypothesis 16 (H16).** *The effect of MGC on tourists' intentions to use social media for planning travel to Saudi Arabia will differ based on whether they have made previous visits.*

**Hypothesis 17 (H17).** *The effect of UGC on tourists' intentions to use social media for planning travel to Saudi Arabia will differ based on whether they have made previous visits.*

**Hypothesis 18 (H18).** *The effect of MGC on tourists' intentions to use social media for planning travel to Saudi Arabia will differ by season.*

**Hypothesis 19 (H19).** *The effect of UGC on tourists' intentions to use social media for planning travel to Saudi Arabia will differ by season.*

Furthermore, the length of stay can also influence a tourist's use of MGC and UGC on social media. Tourists who plan to stay longer may be more likely to use MGC and UGC to research different places and activities [43]. The accommodation type can also influence a tourist's use of MGC and UGC on social media [43]. Tourists who plan to stay in hotels may be more likely to use MGC. In contrast, tourists who plan to stay in hostels or other budget accommodations may be more likely to use UGC. Consequently, this study proposes the following hypotheses:

**Hypothesis 20 (H20).** *The effect of MGC on tourists' intentions to use social media for planning travel to Saudi Arabia will differ based on the length of stay.*

**Hypothesis 21 (H21).** *The effect of UGC on tourists' intentions to use social media for planning travel to Saudi Arabia will differ based on the length of stay.*

**Hypothesis 22 (H22).** *The effect of MGC on tourists' intentions to use social media for planning travel to Saudi Arabia will differ based on the type of accommodation.*

**Hypothesis 23 (H23).** *The effect of UGC on tourists' intentions to use social media for planning travel to Saudi Arabia will differ based on the type of accommodation.*

Overall, this study suggests that tourists' intentions to use social media platforms for planning travel to Saudi Arabia are influenced by PEOU, PU, SAT, MGC, and UGC. The characteristics of tourists and the characteristics of the visit influence the tourists' intentions to use MGC and UGC on social media platforms for planning travel to Saudi Arabia.

**4. Research Methodology**

Previous studies in the literature on tourists' acceptance of using social media as a travel planning tool and the impact of social media content on tourists' travel decisions formed a cornerstone of this thesis. This study adopted the Design Science Research (DSR) methodology as a research approach to develop an artifact, which is a tourist-based framework for developing digital marketing for SMEs in the tourism sector in Saudi Arabia [44].

As a suggestion to deal with the gap of this study, a tourist-based theoretical framework was proposed to assist Saudi tourism SMEs in developing an effective digital marketing strategy on social media. While the purpose of this study is not to formulate a new theory, the TAM was used as a theoretical framework to develop and validate this study's hypotheses, as detailed in Section 2. Then, the tourist-based ML classification model was developed to assist Saudi tourism SMEs in developing an effective digital marketing strategy on social media based on segments of tourists who prefer MGC or UGC when using social media for planning travel to Saudi Arabia. First, the hypotheses of this study were developed to identify the variables that influence tourists' intentions and decisions while using social media to plan travel to Saudi Arabia. Then, the target tourists' primary data were collected to test the tourist-based theoretical framework and develop the ML model.

Based on the literature and previous studies, there is a lack of research on social media marketing for Saudi tourism SMEs, so no preliminary data can be used. As a result, a primary data collection method was used for this study to collect data from the target population. In this study, the target population is tourists worldwide who intend to visit or re-visit Saudi Arabia and use social media to plan their trip. The tourists in the sample may be considered potential customers who can be effectively segmented and targeted.

Since it is difficult to include everyone in the population as participants in a study, it is essential to select a sample that is representative of the target population and large enough to answer the research questions [44]. As a result, a convenience non-probability sampling technique was adopted to recruit a diverse sample of tourists visiting the study destination. However, it is important to acknowledge that convenience samples are not inherently representative of the target population. To mitigate this limitation, the participants were from a variety of different countries to ensure that the sample was diverse in terms of ethnicity, age, and other demographic characteristics. To avoid bias, the sample of tourists enrolling should be just a particular slice of the tourists who reflect the target population being studied. A standardized recruitment procedure was used to ensure that all potential participants were given an equal opportunity to participate in this study.

### 4.1. Study Instruments

A questionnaire was designed and used as the data collection instrument to collect primary data for this study. The questionnaire was created based on constructs and measures from previous studies that used TAM measures in fields such as information systems, social networking, social media, marketing, and tourism literature. Five items were used alongside IU [32,34]. Two items were used to measure PEOU [32]. Similarly, two items were used to measure PU [34]. Tourists' satisfaction with social media content was measured using six items [36]. A description of each measurement item can be found in Table 1.

Moreover, ten items were adopted to measure MGC and ten items to measure UGC [34]. The items for MGC were designed to assess the intention to use MGC, the usefulness of MGC, the ease of using MGC, and tourist satisfaction with MGC. The items for UGC were designed to assess the intention to use UGC, the usefulness of UGC, the ease of using UGC, and tourist satisfaction with UGC. A description of each measurement item can be found in Table 2.

The questionnaire was divided into four sections. The first section consisted of nine questions to obtain the demographic data of tourists, consisting of gender, age, education level, monthly income, and nationality. The questions on visit characteristics included the previous visit, length of stay, type of accommodation, and preferred season. Also, to decide which respondents fulfilled the study's criteria, questions about using social media for travel planning and their intention to visit/re-visit Saudi Arabia were included as filter questions in the first section of the questionnaire.

The second section consisted of nine questions designed to measure the degree of tourists' agreement and disagreement with IU, PU, and PEOU for using social media to plan travel to Saudi Arabia, as well as six questions to assess tourists' satisfaction with the tourism-related content of historic cities, entertainment venues, activities and adventures, hotels, restaurants, and shopping in Saudi Arabia.

The third section consisted of four questions designed to measure the degree of tourists' agreement and disagreement with IU, PU, and PEOU for MGC while using social media to plan travel to Saudi Arabia, as well as six questions to assess tourists' satisfaction with the tourism-related MGC of historic cities, entertainment venues, activities and adventures, hotels, restaurants, and shopping in Saudi Arabia.

**Table 1.** A description of measurement items.

| Constructs | | Items | References |
|---|---|---|---|
| Intention to use SM for travel planning for a visit to Saudi Arabia | IU | Recent social media posts positively affected my desire when planning a trip to Saudi Arabia. | [32,34] |
| | IU. 1 | When planning a trip to Saudi Arabia, I look for restaurants, tour companies, and accommodation profiles on social media. | |
| | IU. 2 | The content shared by restaurants, tour companies, and accommodations on social media influences my desire when planning a trip to Saudi Arabia. | |
| | IU. 3 | Photos, videos, recommendations, and reviews on social media influence my desire when planning a trip to Saudi Arabia. | |
| | IU. 4 | I rely on users' photos, videos, and comments on social media to make decisions when planning a trip to Saudi Arabia. | |
| Perceived Usefulness (PU) | PU | I consider the content on the profiles of restaurants, tour companies, and accommodations on social media to be helpful information when planning a trip to Saudi Arabia. | [34] |
| | PU. 1 | I consider comments, photos, and videos which travelers share on social media to be a useful source of information when planning a trip to Saudi Arabia. | |
| Perceived Ease of Use (PEOU) | PEOU | It is easy to reach profiles of restaurants, tour companies, and accommodations on social media when planning a trip to Saudi Arabia. | [32] |
| | PEOU. 1 | Obtaining traveler recommendations and reviews about travel to Saudi Arabia is easy. | |
| Satisfaction (SAT) | SAT | How satisfied are you with content on social media about historic cities in Saudi Arabia? | [36] |
| | SAT. 1 | How satisfied are you with content on social media about entertainment venues in Saudi Arabia? | |
| | SAT. 2 | How satisfied are you with content on social media about activities and adventures in Saudi Arabia? | |
| | SAT. 3 | How satisfied are you with content on social media about hotels in Saudi Arabia? | |
| | SAT. 4 | How satisfied are you with content on social media about restaurants in Saudi Arabia? | |
| | SAT. 5 | How satisfied are you with content on social media about shopping in Saudi Arabia? | |

The last section of the questionnaire contained four questions designed to measure the degree of tourists' agreement and disagreement with IU, PU, and PEOU for UGC while using social media to plan travel to Saudi Arabia, as well as six questions to assess tourists' satisfaction with the tourism-related UGC of historic cities, entertainment venues, activities and adventures, hotels, restaurants, and shopping in Saudi Arabia.

A closed-ended question format was adopted in all sections of the questionnaire, with each participant choosing one or more responses from a list of multiple options. Closed-ended questions are simpler to use, more helpful in the comparison analysis, and minimize subjective bias in the results [45]. In the first section of the questionnaire, a nominal scale was used to measure demographic data for each tourist and visit characteristic, and a yes/no scale was used in the filter questions. A five-point Likert scale was applied in the questionnaire's second, third, and fourth sections to measure the attitude score, in which 1 means "strongly agree" and 5 means "strongly disagree". A "Neutral" degree is given as 3.

<p align="center">**Table 2.** A description of measurement items of MGC and UGC.</p>

| Constructs | | Items | References |
|---|---|---|---|
| Marketer-Generated Content (MGC) | MGC | When planning a trip to Saudi Arabia, I look for MGC of restaurants, tour companies, and accommodation profiles on social media. | [34] |
| | MGC. 1 | The MGC shared by restaurants, tour companies, and accommodations on social media influences my desire when planning a trip to Saudi Arabia. | |
| | MGC. 2 | I consider the MGC on the profiles of restaurants, tour companies, and accommodations on social media to be helpful information when planning a trip to Saudi Arabia. | |
| | MGC. 3 | It is easy to reach the MGC on social media profiles of restaurants, tour companies, and accommodations when planning a trip to Saudi Arabia. | |
| | MGC. 4 | How satisfied are you with MGC on social media about historic cities in Saudi Arabia? | |
| | MGC. 5 | How satisfied are you with MGC on social media about entertainment venues in Saudi Arabia? | |
| | MGC. 6 | How satisfied are you with MGC on social media about activities and adventures in Saudi Arabia? | |
| | MGC. 7 | How satisfied are you with MGC on social media about hotels in Saudi Arabia? | |
| | MGC. 8 | How satisfied are you with MGC on social media about restaurants in Saudi Arabia? | |
| | MGC. 9 | How satisfied are you with MGC on social media about shopping in Saudi Arabia? | |
| User-Generated Content (UGC) | UGC | UGC such as photos, videos, recommendations, and reviews on social media influence my desire when planning a trip to Saudi Arabia. | [34] |
| | UGC. 1 | I rely on UGC such as users' photos, videos, and comments on social media to make decisions when planning a trip to Saudi Arabia. | |
| | UGC. 2 | I consider UGC such as comments, photos, and videos that travelers share on social media to be helpful information when planning a trip to Saudi Arabia. | |
| | UGC. 3 | Reaching UGC such as traveler recommendations and reviews about travel to Saudi Arabia is easy. | |
| | UGC. 4 | How satisfied are you with UGC on social media about historic cities in Saudi Arabia? | |
| | UGC. 5 | How satisfied are you with UGC on social media about entertainment venues in Saudi Arabia? | |
| | UGC. 6 | How satisfied are you with UGC on social media about activities and adventures in Saudi Arabia? | |
| | UGC. 7 | How satisfied are you with UGC on social media about hotels in Saudi Arabia? | |
| | UGC. 8 | How satisfied are you with UGC on social media about restaurants in Saudi Arabia? | |
| | UGC. 9 | How satisfied are you with UGC on social media about shopping in Saudi Arabia? | |

A random sample of 10 tourists from the target population of tourists who intended to visit/re-visit Saudi Arabia were recruited to participate in the face and content validation of the questionnaire. The sample was selected from multiple age groups, genders, and regions to ensure representativeness. Face and content validity were assessed by a panel of three experts in IS and tourism research. The experts reviewed the questionnaire items for accuracy, understandability, and relevance to the research objectives. A small number of the questions were modified after the pilot test. Microsoft Forms was used to conduct the questionnaire to expedite the data collection since it is a simple and valuable tool for

quickly collecting data from many target participants. The responses submitted to the link were kept private and password-protected.

Since the target population must use social media, reaching them through social media platforms is simple. The survey was distributed via email, Facebook, Twitter, and Instagram for two months, between 20 August and 20 October 2022. The survey was distributed to tourists, and 710 completed responses were collected. As this study focused on tourists who intended to use social media to plan travel to Saudi Arabia, 137 responses that did not adhere to the study's criteria were excluded from the analysis, including 53 responses from tourists who did not use social media to plan their travels and 84 responses from tourists who did not intend to visit Saudi Arabia, resulting in a total of 573 complete, valid responses.

### 4.2. Data Cleaning and Encoding

To ensure the accuracy of our study, we took several steps. First, we removed the "id" column, as it was irrelevant. We then thoroughly validated the dataset and found no errors or missing data, since all survey questions were closed-ended, and each was designated as a required field to be filled out to submit the form. The columns containing timestamps were removed to protect the participants' privacy. Also, the filter question columns were eliminated, as they all had the same response. Then, the responses that were not compliant with the criteria for inclusion were removed; 137 out of 710 responses were rejected from the analysis, including 53 responses from tourists who do not use social media for travel planning and 84 from tourists who do not intend to visit Saudi Arabia. Our dataset consisted of 573 completed responses with 15 variables in 44 columns.

### 4.3. Data Analysis and Hypothesis Testing

This study's data analysis and hypothesis testing were conducted using Jupyter Notebook and the Python 3.6 programming language. Python offers a range of tools for analyzing data, gaining insights, testing hypotheses, and making informed decisions to develop ML models. The dataset was analyzed using a descriptive analysis, reliability testing, and a correlation coefficient analysis. The descriptive analysis uses the mean and standard deviation to summarize and describe the data's critical characteristics.

Reliability testing was carried out to examine the reliability of the measurement instrument, using Cronbach's alpha values to estimate the internal consistency of the items of each construct. An acceptable consistency coefficient value should be at least 0.70 [46]. A Cronbach's alpha value of more than 0.70 indicates that all constructs were reliable and appropriate for inclusion in the final model [46]. Lastly, the correlation coefficient, a statistical technique for exploring the relationship between variables, was also used.

The hypotheses of the tourist-based theoretical framework were tested using ordinary least squares (OLS) regression to assess the relationship between the dependent, independent, and moderator variables [47]. The IUs of social media platforms for travel planning were defined as dependent variables. The categorical independent variables of PU, PEOU, SAT, MGC and UGC were identified as independent variables. The categorical moderator variables included the tourists' characteristics (gender, age, education level, monthly income, and nationality) and visit characteristics (previous visit, length of stay, type of accommodation, and travel party composition).

These variables were used to test whether the relationship between the independent and dependent variables differed across different categories of moderator variables. A composite score for each construct of the tourist-based theoretical framework was created to accomplish this, followed by the interaction terms between independent and moderator variables.

Finally, the regression model was fitted and the regression results were printed using the "summary" function. The output of the "summary" function includes the coefficients, standard errors, t-values, and *p*-values for each variable in the model, including the interaction terms. The moderator variable interaction effect was interpreted for each category by

examining the coefficient of the interaction term for that category, representing the slope difference between the independent and dependent variables for each unit increase in independent variables [48]. A significant *p*-value of the interaction term for a particular category indicates that the interaction impact is statistically significant for that category [48].

*4.4. Tourist-Based ML Model Development*

Developing the tourist-based ML classification model, which segments tourists into two groups based on their preference for MGC and UGC, involves several steps, including data splitting, algorithm selection, classification model training, and selecting the best model.

The first step was data splitting, as the dataset was cleaned and encoded. In this study, we utilized a splitting ratio of 70/30, where 70% of the data is used for training and 30% for testing the model. Next, seven ML classification algorithms are selected to train and test the tourist-based ML model. Further, various tree-based classification algorithms are used, including the decision tree classifier, extra tree classifier, and random forest classifier. The decision tree and extra tree classifiers are simple to understand and interpret and can handle non-linear relationships between features.

The precision score reflects the proportion of true positives among all predicted positives, while the recall score reflects the proportion of true positives among all actual positives. The f1-score is a calculation that combines precision and recall, while support refers to the number of instances in each class.

*4.5. Evaluation*

The evaluation phase is the process of measuring the efficiency of the model performance. This study used the K-Fold Cross-Validation (KF-CV) technique to evaluate the performance of the tourist-based classification model. K-fold involves splitting the data into K equal-sized subsets or folds [49]. The model is then trained on K-1 folds of the data and tested on the remaining fold. This process is repeated K times, with each fold as the test set once. The results of each test are then averaged to obtain a final score for the model's performance. Using K-fold helps to ensure that the model's performance is not biased based on the specific subset of data used for training and testing.

## 5. Data Analysis and Results

*5.1. Descriptive Analysis*

5.1.1. Respondents' Demographic Profile

The survey received 573 responses, with 433 (75.56%) male and 140 (24.43%) female respondents. According to the results by age group, we found that the largest age group among our participants was 245 between the ages of 26 and 35, making up a significant (42.75%) proportion of the total sample. The 36–45 age range ranked second, accounting for 110 (19.19%) of the respondents. The 18–25 age group also had a notable presence, comprising 179 (31.23%) of the participants. On the other hand, the remaining age groups, i.e., 46–55, under 18, and 56–65, had relatively fewer frequencies and percentages, with 20 (3.49%), 12 (2.09%), and seven (1.22%), respectively. It is crucial to note that none of our survey respondents were over 65. These results provide valuable insight into the age demographics of our target audience, which we can use to tailor our marketing and outreach strategies accordingly. The gender and age groups of the tourists are illustrated in Figure 2.

The analysis of the respondents' nationality showed that most were Americans, representing 365 (63.69%) of the total sample. The other respondents came from various countries, with different frequencies and percentages. The second most common nationality was Egyptian, with 40 (6.98%) respondents, followed by Pakistani, with 21 (3.66%) respondents. The other nationality groups that had more than 10 respondents each were Yemeni, with 16 (2.79%) respondents, Emirati, with 15 (2.61%), French (14; 2.44%), British (12; 2.09%), and Italian (12; 2.09%). The least common nationalities were Indian and Cana-

dian, with only 11 (1.91%) and nine (1.57%) respondents, respectively. The remaining 58 (10.12%) were from various nationalities. The nationalities of the tourists are illustrated in Figure 3.

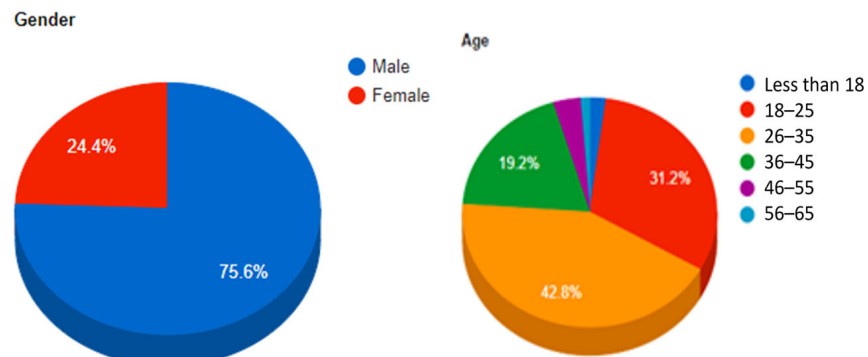

**Figure 2.** The gender and age groups of the tourists (source: created by the authors).

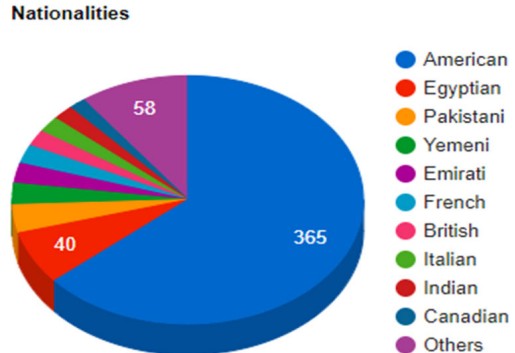

**Figure 3.** The nationalities of the tourists (source: created by the authors).

The educational level and monthly income of the respondents were also analyzed in the survey. The results revealed that the most prevalent educational level was a bachelor's degree, with 257 (44.85%) respondents holding this qualification. The second most prevalent educational level was a professional diploma degree, with 228 respondents (39.79%) having this qualification. A smaller proportion of the respondents had a high-school certificate or less, with 55 (9.59%) respondents falling into this category. The least prevalent educational level was a postgraduate degree, with only 33 (5.75%) respondents possessing this qualification.

The respondents' monthly income results, divided into five categories, ranging from less than SAR 5000 to more than SAR 30,000, indicated that the most common income category among the respondents was SAR 20,001–30,000, with 185 (32.28%) respondents reporting this income level. The second most common income category was above SAR 30,000, with 131 (22.86%) respondents reporting this income level. The third most common income category was SAR 10,001–20,000, with 113 (19.72%) respondents reporting this income level. The fourth most common income category was SAR 5001–10,000, with 88 (15.35%) respondents reporting this income level. The least common income category was below SAR 5000, with only 56 (9.77%) respondents reporting this income level. The education level and monthly income of tourists are illustrated in Figure 4.

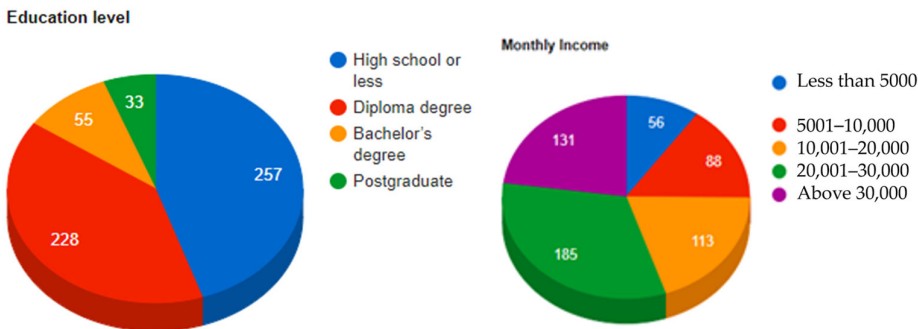

**Figure 4.** Education level and monthly income (source: created by the authors).

The demographic characteristics of the tourists who participated in the questionnaire survey are presented in Table 3.

**Table 3.** Demographic characteristics.

| Tourists' Characteristics | Category | Frequency (n) | (%) |
|---|---|---|---|
| Gender | Male | 433 | (75.56%) |
| | Female | 140 | (24.43%) |
| Age | Less than 18 | 12 | (2.09%) |
| | 18–25 | 179 | (31.23%) |
| | 26–35 | 245 | (42.75%) |
| | 36–45 | 110 | (19.19%) |
| | 46–55 | 20 | (3.49%) |
| | 56–65 | 7 | (1.22%) |
| Nationality | American | 365 | (63.69%) |
| | Egyptian | 40 | (6.98%) |
| | Pakistani | 21 | (3.66%) |
| | Yemeni | 16 | (2.79%) |
| | Emirati | 15 | (2.61%) |
| | French | 14 | (2.44%) |
| | British | 12 | (2.09%) |
| | Italian | 12 | (2.09%) |
| | Indian | 11 | (1.91%) |
| | Canadian | 9 | (1.57%) |
| | Others | 58 | (10.12%) |
| Education level | High school or less | 257 | (44.85%) |
| | Professional diploma degree | 228 | (39.79%) |
| | Bachelor's degree | 55 | (9.59%) |
| | Postgraduate | 33 | (5.75%) |
| Monthly income | Less than SAR 5000 | 56 | (9.77%) |
| | SAR 5001–SAR 10,000 | 88 | (15.35%) |
| | SAR 10,001–SAR 20,000 | 113 | (19.72%) |
| | SAR 20,001–SAR 30,000 | 185 | (32.28%) |
| | More than SAR 30,000 | 131 | (22.86%) |

### 5.1.2. Visit Characteristics

The analysis of the data collected on the visit characteristics showed that 386 respondents (63.27%) had not previously visited Saudi Arabia, while 187 (30.65%) had experienced at least one visit. In terms of the duration of stay, the survey found that the majority of respondents preferred a seven-day stay, comprising 286 (46.88%) of the sample. Following closely behind was the fourteen-day stay, with 228 (37.37%) of the respondents. The other durations of stay, i.e., twenty-one days, twenty-nine days, and over a month, were less popular, with only 32 (5.24%), 18 (2.95%), and nine (1.47%), respectively. The previous visit and duration-of-stay findings are illustrated in Figure 5.

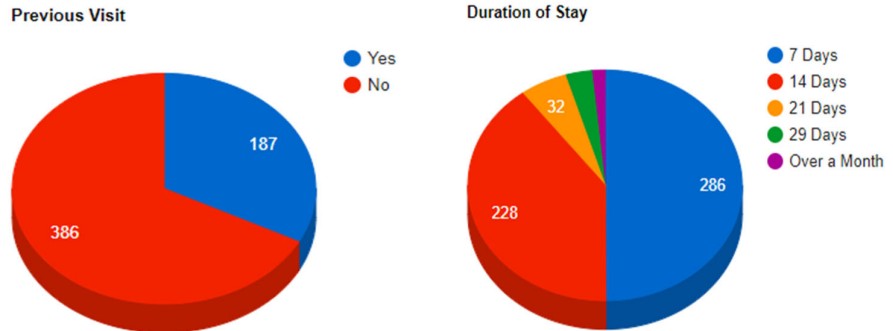

**Figure 5.** Previous visit and duration of stay (source: created by the authors).

This study also explored the preferred types of accommodation. It was found that economy hotels or motels were the most favored option, with 291 (47.7%) respondents opting for them. Luxury hotels (four stars and above) were also popular, with 157 (25.73%) respondents indicating a preference. On the other hand, resorts and holiday houses were less favored, with only 93 (15.24%) and 32 (5.24%) of respondents choosing them, respectively. Moreover, the data analysis indicated that 182 tourists (29.83%) preferred to visit Saudi Arabia in the summer. They were followed by 148 (24.26) tourists who preferred winter, 132 (21.63%) tourists who preferred spring, and 111 (18.19%) tourists who preferred autumn. The type of accommodation and preferred season are illustrated in Figure 6.

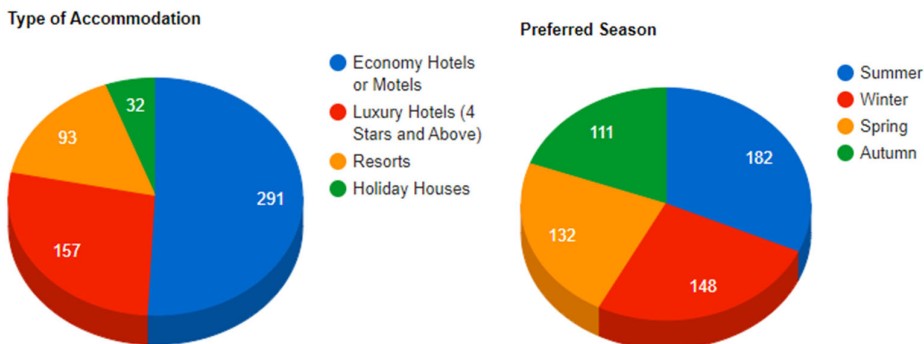

**Figure 6.** The type of accommodation and preferred season (source: created by the authors).

The frequency and percentage of the visit characteristic data of the tourist respondents are shown in Table 4.

**Table 4.** Visit characteristics.

| Visit Characteristic | Category | Frequency (n) | (%) |
|---|---|---|---|
| Previous Visit | Yes | 187 | (30.65%) |
| | No | 386 | (63.27%) |
| Duration of Stay | 7 Days | 286 | (46.88%) |
| | 14 Days | 228 | (37.37%) |
| | 21 Days | 32 | (5.24%) |
| | 29 Days | 18 | (2.95%) |
| | Over a Month | 9 | (1.47%) |
| Type of Accommodation | Economy Hotels or Motels | 291 | (47.70%) |
| | Luxury Hotels (4 Stars and Above) | 157 | (25.73%) |
| | Resorts | 93 | (15.24%) |
| | Holiday Houses | 32 | (5.24%) |
| Preferred Season | Summer | 182 | (29.83%) |
| | Winter | 148 | (24.26%) |
| | Spring | 132 | (21.63%) |
| | Autumn | 111 | (18.19%) |

*5.2. Descriptive Statistics of Constructs*

Descriptive statistics summarize and present the characteristics of the dataset, such as the mean, standard deviation, and frequency of each variable relevant [50]. In this study, the six constructs—IU, PU, PEOU, SAT, MGC, and UGC—are measured based on the mean and standard deviation (SD) by averaging the scores of the items that measured them on the Likert scale. The results are shown in Table 5.

**Table 5.** Descriptive statistics of constructs.

| Construct | Mean | SD | N |
|-----------|------|------|-----|
| IU | 3.62 | 1.39 | 573 |
| PU | 3.71 | 1.25 | 573 |
| PEOU | 3.69 | 1.32 | 573 |
| SAT | 4.13 | 0.78 | 573 |
| MGC | 3.69 | 1.25 | 573 |
| UGC | 3.96 | 1.06 | 573 |

The table shows that the mean scores of all constructs are above the midpoint of the scale (three), indicating that the respondents generally have positive attitudes toward the variables. The highest mean score is for SAT (4.13), followed by UGC (3.96), PU (3.71), PEOU and MGC (both 3.69), and IU (3.62). The SD measures the variability or dispersion of the scores around the mean. The lowest standard deviation is for SAT (0.78), indicating that the scores are more consistent and less spread out than the other constructs. The highest standard deviation is for IU (1.39), indicating that the scores are more diverse and less consistent than the other constructs. The SDs of PU, PEOU, UGC, and MGC are similar, ranging from 1.06 to 1.32.

*5.3. Reliability Analysis*

Reliability analysis is a fundamental statistical technique that is commonly used to evaluate the quality and consistency of measures or scales employed in assessing a particular concept [49]. In this study, we used Cronbach's alpha to measure the internal consistency of six constructs: IU, PU, PEOU, SAT, MGC, and UGC. Multiple items on a Likert scale were employed to evaluate each construct, and the Cronbach's alpha coefficient was calculated for each construct. As presented in Table 6, our study's results indicate high Cronbach's alpha coefficients for all six constructs, signifying consistent and reliable measurements across the board. Notably, the highest Cronbach's alpha coefficient is for MGC (0.935), followed by IU (0.908), SAT (0.860), PEOU (0.786), UGC (0.754), and PU (0.735). Consequently, our findings suggest that the measures and scales employed in this study are reliable and consistent in assessing the targeted constructs. Therefore, researchers and practitioners can confidently utilize these measures and scales to assess the constructs of interest.

**Table 6.** Cronbach's alpha coefficients of constructs.

| Construct | Cronbach's Alpha |
|-----------|------------------|
| IU | 0.908 |
| PU | 0.735 |
| PEOU | 0.786 |
| SAT | 0.860 |
| MGC | 0.935 |
| UGC | 0.754 |

*5.4. Correlation Analysis*

Correlation analysis is a valuable statistical tool for determining the strength and direction of the linear relationship between two variables, which also enables the evaluation of their association [51]. A correlation coefficient between −1 and 1 is typically used to determine the strength and direction of the relationship between the variables [51]. In this sense, a correlation coefficient of 1 signifies a perfect positive correlation, while a correlation coefficient of −1 indicates a perfect negative correlation. On the other hand, a correlation coefficient of 0 suggests no linear association between the variables. The Spearman correlation coefficient is a nonparametric method that gauges the rank correlation and does not assume a normal data distribution [51]. This study used the Spearman correlation coefficient to assess the relationships between the variables. The outcomes of the correlation analysis between variables are presented in Table 7.

**Table 7.** The correlation analysis of constructs.

|      | IU   | PU   | PEOU | SAT  | MGC  | UGC  |
|------|------|------|------|------|------|------|
| IU   | 1.0  | 0.86 | 0.87 | 0.86 | 0.92 | 0.76 |
| PU   | 0.86 | 1.0  | 1.0  | 0.86 | 0.96 | 0.83 |
| PEOU | 0.87 | 1.0  | 1.0  | 0.86 | 0.96 | 0.83 |
| SAT  | 0.86 | 0.86 | 0.86 | 1.0  | 0.90 | 0.96 |
| MGC  | 0.92 | 0.96 | 0.96 | 0.90 | 1.0  | 0.83 |
| UGC  | 0.76 | 0.83 | 0.83 | 0.93 | 0.83 | 1.0  |

The Spearman correlation coefficient between IU and PU, PEOU, SAT, MGC, and UGC showed that IU had a strong positive correlation with MGC (r = 0.920), a strong positive correlation with PU (r = 0.863), POUE (r = 0.870), and SAT (r = 0.863), and a moderate positive correlation with UGC (r = 0.759). These results suggest that IU is influenced by all five variables, with MGC being the most influential factor. Likewise, the findings indicate a strong positive correlation between PU and all the other variables. The strongest correlation is between PU and PEOU (r = 0.990), which means that the PU and PEOU of social media platforms are directly related. If the PU is high, the PEOU will also be high, and vice versa. The correlation between PU and IU is also strong and positive (r = 0.863), which means that as the PU of social media platforms increases, so does the IU of the social media platforms to plan travel to Saudi Arabia, and vice versa. The correlation between PU and SAT is also strong and positive (r = 0.858), which indicates that satisfaction with social media platforms increases with their PU of the social media platforms to plan travel to Saudi Arabia, and vice versa. Furthermore, there is a strong and positive correlation (r = 0.958) between PU and MGC. This implies that the greater the PU of social media platforms, the greater the motivation to use MGC on the social media platforms to plan travel to Saudi Arabia, and vice versa. The weakest correlation is between PU and UGC (r = 0.834), which means that the PU of social media platforms has a lesser impact than the other variables on the motivation to use UGC on the social media platforms to plan travel to Saudi Arabia.

According to the results, there is also a strong positive correlation between PEOU and all other variables. The correlation between PEOU and SAT is also strong and positive (r = 0.859), which means that if social media platforms have a high PEOU, users are more likely to be satisfied with the social media platforms to plan travel to Saudi Arabia. Similarly, if users are satisfied with the platforms, they are more likely to perceive them as easy to use. The correlation between PEOU and MGC is strongly positive (r = 0.964), which means that the more accessible social media platforms are used, the more motivated users are to use MGC on the social media platforms to plan travel to Saudi Arabia. The weakest correlation is between PEOU and UGC (r = 0.829), which means that, although the more user-friendly social media platforms are, the more likely people are to use UGC on social media platforms to plan travel to Saudi Arabia, this relationship is less significant than other factors.

The second strongest correlation is between SAT and UGC (r = 0.925), which is a strong positive correlation. This means that the higher the satisfaction with the social media platforms, the higher the motivation to use UGC on social media platforms to plan travel to Saudi Arabia, and vice versa. The correlation between SAT and IU is also strong and positive (r = 0.863), which means that the higher the satisfaction with social media platforms, the higher the intention to use social media platforms to plan travel to Saudi Arabia, and vice versa. The correlation between SAT and PEOU is also strong and positive (r = 0.859), which means that the higher the satisfaction with social media platforms, the higher the PEOU of the social media platforms, and vice versa. The correlation between SAT and MGC is strongly positive (r = 0.905), which means that the higher the satisfaction with social media platforms, the higher the motivation to use MGC on the social media platforms to plan travel to Saudi Arabia, and vice versa. The weakest correlation is between SAT and PU (r = 0.858), which means satisfaction with social media platforms positively correlates with PU, but that this relationship is relatively weak compared to other variables.

The results of the correlation analysis of MGC with all variables show that the strongest correlation is between MGC and PEOU (r = 0.964), which indicates that when there is a high motivation to use MGC on social media platforms for planning travel to Saudi Arabia, there is also a high PEOU of those platforms for the same purpose, and vice versa. The correlation between MGC and IU is also strong and positive (r = 0.920), which means that the motivation to use MGC on social media platforms increases, so does the use of social media platforms to plan travel to Saudi Arabia, and vice versa. There is a strong positive correlation (r = 0.905) between the motivation to use MGC on social media for planning trips to Saudi Arabia and the satisfaction with social media platforms for such planning. This means that as the motivation to use MGC increases, so does the satisfaction with social media platforms for trip planning, and vice versa. The correlation between MGC and PU is also strong and positive (r = 0.958), which means that if a tourist is highly motivated to use MGC for planning travel to Saudi Arabia on social media platforms, the tourist is likely to have a higher PU of those platforms for travel planning purposes, and vice versa. The weakest correlation is between MGC and UGC (r = 0.833), which is still a strong positive correlation. This means that the level of motivation to use MGC or UGC on social media platforms for planning travel to Saudi Arabia is directly proportional, but this relationship is weaker than with other variables.

The correlation analysis of UGC with all variables shows that the strongest correlation is between UGC and SAT (r = 0.925), which is a strong positive correlation. This means that the higher the motivation to use UGC on social media platforms to plan travel to Saudi Arabia, the higher the satisfaction with the social media platforms, and vice versa. The correlation between UGC and PU is also strong and positive (r = 0.834), which implies that the more motivated a tourist is to use UGC on social media platforms to plan their travel to Saudi Arabia, the higher their PU of the social media platforms will be, and vice versa. The correlation between UGC and PEOU is also strong and positive (r = 0.829), which indicates that higher motivation to use UGC on social media platforms to plan travel to Saudi Arabia leads to higher PEOU of social media platforms, and vice versa. The correlation between UGC and MGC is also strong and positive (r = 0.833), indicating a that greater desire to use UGC on social media platforms for planning travel to Saudi Arabia is associated with a greater desire to use MGC, and vice versa. The weakest correlation is between UGC and IU (r = 0.759), which is still a strong positive correlation. This means that the higher the motivation to use UGC on social media platforms for planning travel to Saudi Arabia, the greater the IU for social media platforms, but that this relationship is less significant compared to other factors.

A heatmap illustrating the results of the correlation analysis is shown in Figure 7.

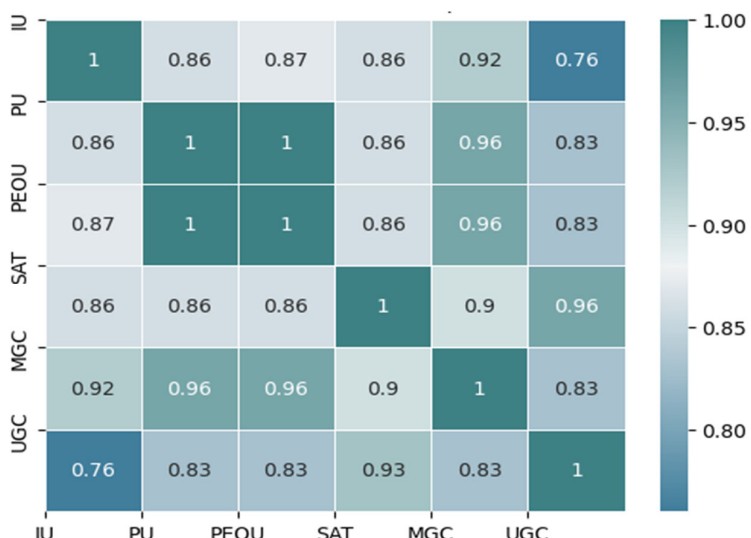

**Figure 7.** Heatmap of correlation analysis (source: created by the authors).

*5.5. Hypothesis Testing*

This study investigated the behavior of tourists on social media by identifying their utilization of social media as a new technology when planning to travel to Saudi Arabia and determining the factors that influence their decisions. The tourist-based theoretical framework hypotheses were developed and tested with OLS to achieve this study's objectives.

5.5.1. The Influence of PU, PEOU, SAT, MGC, and UGC

This study suggested that tourists' intentions to use social media platforms for planning travel to Saudi Arabia are influenced by all the variables, PEOU, PU, SAT, MGC, and UGC.

The OLS results of **H1**: *PU positively affects tourists' intentions to use social media for planning travel to Saudi Arabia.* The OLS results give a coefficient of 0.9706, with a standard error of 0.019, a t-value of −4.279, and a *p*-value of 0.000011. Since the *p*-value is less than 0.05, the null hypothesis can be rejected, and the alternative hypothesis can be accepted. Therefore, the results support **H1**, which means that PU positively impacts tourists' intentions to use social media for planning travel to Saudi Arabia. In simple terms, tourists are more likely to use social media for travel planning if they find it useful.

The OLS results of **H2**: *PEOU positively affects tourists' intentions to use social media for planning travel to Saudi Arabia*, since the coefficient is 0.9086, with a standard error of 0.016, a t-value of −3.294, and a *p*-value of 0.000524. Since the *p*-value is less than 0.05, the null hypothesis can be rejected, and the alternative hypothesis can be accepted. Therefore, the results support **H2**, which means that PEOU positively impacts tourists' intentions to use social media for planning travel to Saudi Arabia. In simple terms, tourists are more likely to use social media for travel planning if they find it easy to use.

The OLS results of **H3**: *SAT positively affects tourists' intentions to use social media for planning travel to Saudi Arabia*, since the coefficient is 1.7337, with a standard error of 0.041, a t-value of 4.190, and a *p*-value of 0.000016. Since the *p*-value is less than 0.05, the null hypothesis can be rejected, and the alternative hypothesis can be accepted. Therefore, the results support **H3**, which means that SAT positively impacts tourists' intentions to use social media for planning travel to Saudi Arabia. In simple terms, tourists are more likely to use social media for travel planning if they are satisfied with the content on social media platforms.

The OLS results of **H4**: *MGC positively affects tourists' intentions to use social media for planning travel to Saudi Arabia*, since the coefficient is 1.1389, with a standard error of 0.016, a t-value of −3.966, and a *p*-value of 0.000041. Since the *p*-value is less than 0.05, the null hypothesis can be rejected, and the alternative hypothesis can be accepted. Therefore,

the results support **H4**, which means that MGC positively impacts tourists' intentions to use social media for planning travel to Saudi Arabia. In simple terms, tourists are more likely to use social media for travel planning if they are satisfied with MGC on social media platforms.

The OLS results of **H5**: *UGC positively affects tourists' intentions to use social media for planning travel to Saudi Arabia*, since the coefficient is 1.6279, with a standard error of 0.053, a t-value of 3.946, and a *p*-value of 0.000045. Since the *p*-value is less than 0.05, the null hypothesis can be rejected, and the alternative hypothesis can be accepted. Therefore, the results support **H5**, which means that UGC positively impacts tourists' intentions to use social media for planning travel to Saudi Arabia. In simple terms, tourists are more likely to use social media for travel planning if they are satisfied with UGC on social media platforms.

Based on the hypothesis testing conducted, it was found that tourists have a positive attitude towards using social media for planning travel in Saudi Arabia. This is mainly due to the user-friendly and useful features provided by various social media platforms. Also, the content related to Saudi Arabia's tourism on social media, whether MGC or UGC, significantly impacts tourists' decision-making processes, ultimately leading to a better travel experience. The results of testing hypotheses **H1** to **H5** are presented in Table 8.

**Table 8.** Hypothesis testing from **H1** to **H5**.

| Hypothesis | Coefficient | Std Err | t-Value | *p*-Value | Result |
|:---:|:---:|:---:|:---:|:---:|:---:|
| **H1** | 0.9086 | 0.016 | −4.279 | 0.000011 | Reject the null hypothesis |
| **H2** | 0.9706 | 0.019 | −3.294 | 0.000524 | Reject the null hypothesis |
| **H3** | 1.7337 | 0.041 | 4.190 | 0.000016 | Reject the null hypothesis |
| **H4** | 1.1389 | 0.016 | −3.966. | 0.000041 | Reject the null hypothesis |
| **H5** | 1.6279 | 0.053 | 3.946 | 0.000045 | Reject the null hypothesis |

5.5.2. The Influence of the Tourists' Characteristics

This study delves into the impact of MGC and UGC on the intention of tourists to use social media platforms for planning travel to Saudi Arabia. To examine how tourist characteristics affect their intention to use social media to plan travel to Saudi Arabia, the hypotheses were tested using OLS.

**H6** and **H7** examine how MGC and UGC influence the intention of tourists, based on their gender, to use social media for planning travel to Saudi Arabia. The OLS results of "**H6**: *The effect of MGC on tourists' intentions to use social media for planning travel to Saudi Arabia will differ by gender*", reveal that the coefficient is 0.3620, with a standard error of 0.075, a t-value of 4.848, and a *p*-value of 0.000031, indicating a significant effect of gender on the impact of MGC, meaning that male tourists are more likely to be influenced by MGC than female tourists when making travel plans to Saudi Arabia. On the other hand, the OLS results of "**H7**: *The effect of UGC on tourists' intentions to use social media for planning travel to Saudi Arabia will differ by gender*" demonstrate a coefficient of 0.1719, with a standard error of 0.102, a t-value of 1.685, and a *p*-value 0.0894884, indicating no significant effect of gender on the impact of UGC, meaning that there is no significant difference in the way that male and female tourists are influenced by UGC when making travel plans to Saudi Arabia. The results from the OLS analysis indicated that gender affects the influence of MGC, but not UGC, regarding tourists' intentions to use social media for planning travel to Saudi Arabia. As a result, **H6** is accepted, while **H7** is rejected.

**H8** and **H9** examine how MGC and UGC influence the intention of tourists, based on their age, to use social media for planning travel to Saudi Arabia. The OLS results of "**H8**: *The effect of MGC on tourists' intentions to use social media for planning travel to Saudi Arabia will differ by age*", reveal a coefficient of 0.1842, with a standard error of 0.038, a t-value of 4.784, and a *p*-value of 0.000397, indicating a significant effect of age on the impact of MGC. Meanwhile, the OLS results of "**H9**: *The effect of UGC on tourists' intentions to use social media*

*for planning travel to Saudi Arabia will differ by age*", demonstrate that the coefficient is 0.0605, with a standard error of 0.071, a t-value of 0.857, and a *p*-value of 0.000186, indicating a significant effect of age on the impact of UGC. Therefore, **H8** and **H9** are accepted. As shown, the age group analysis results are also interesting. The largest age group among the participants was 245 between the ages of 26 and 35, making up a 42.75% proportion of the total sample. This suggests that younger tourists are a vital target audience for tourism marketing in Saudi Arabia.

The OLS analysis results of hypothesis testing of hypotheses **H6** to **H9** are summarized in Table 9.

**Table 9.** Hypothesis testing from **H6** to **H9**.

| Hypothesis | Coefficient | Std Err | t-Value | *p*-Value | Result |
|:---:|:---:|:---:|:---:|:---:|:---:|
| **H6** | 0.3620 | 0.075 | 4.848 | 0.000031 | Reject the null hypothesis |
| **H7** | 0.1719 | 0.102 | 1.685 | 0.0894884 | Failed to reject the null hypothesis |
| **H8** | 0.1842 | 0.038 | 4.784 | 0.000397 | Reject the null hypothesis |
| **H9** | 0.0605 | 0.071 | 0.857 | 0.000186 | Reject the null hypothesis |

**H10** and **H11** investigate how MGC and UGC influence the intention of tourists, based on their nationality, to use social media for planning travel to Saudi Arabia. The OLS results of "**H10**: *The effect of MGC on tourists' intentions to use social media for planning travel to Saudi Arabia will differ by nationality*", reveal that the coefficient is 0.0536, with a standard error of 0.008, a t-value of 6.505, and a *p*-value of 0.017071, indicating an effect of nationality on the impact of MGC. On the other hand, the OLS results of "**H11**: *The effect of UGC on tourists' intentions to use social media for planning travel to Saudi Arabia will differ by nationality*" demonstrate that the coefficient is 0.0036. with a standard error of 0.016, a t-value of 0.230, and a *p*-value of 0.081823, indicating no significant effect of nationality on the impact of UGC. The results of the OLS showed that nationality influences the impact of MGC but not UGC on tourists' intentions to use social media for planning travel to Saudi Arabia. As a result, **H10** is accepted, while **H11** is rejected.

**H12** and **H13** investigate how MGC and UGC influence the intention of tourists, based on their education level, to use social media for planning travel to Saudi Arabia. The OLS results of "**H12**: *The effect of MGC on tourists' intentions to use social media for planning travel to Saudi Arabia will differ by education level*" reveal that the coefficient is 0.2152, with a standard error of 0.041, a t-value of 5.221, and a *p*-value of 0.002171, indicating a significant positive effect of education level on the impact of MGC. Meanwhile, the OLS results of "**H13**: *The effect of UGC on tourists' intentions to use social media for planning travel to Saudi Arabia will differ by education level*" demonstrate that the coefficient is 0.0362, with a standard error of 0.061, a t-value of 0.593, and a *p*-value of 0.0653, indicating no significant effect of education level on the impact of UGC. Therefore, **H12** is accepted, while **H13** is rejected.

**H14** and **H15** investigate how MGC and UGC influence the intention of tourists, based on their monthly income, to use social media for planning travel to Saudi Arabia. The OLS results of "**H14**: *The effect of MGC on tourists' intentions to use social media for planning travel to Saudi Arabia will differ by monthly income*" reveal a coefficient of 0.2480, with a standard error of 0.014, a t-value of 17.553, and a *p*-value of 0.01912, indicating a significant positive effect of monthly income on the impact of MGC. However, the OLS results of "**H15**: *The effect of UGC on tourists' intentions to use social media for planning travel to Saudi Arabia will differ by monthly income*" demonstrate that the coefficient is 0.0050, with a standard error of 0.041, a t-value of 0.12,1 and a *p*-value of 0.09034, indicating no significant effect of monthly income on the impact of UGC. The results of OLS showed that monthly income influences the impact of MGC but not UGC on tourists' intentions to use social media for planning travel to Saudi Arabia. As a result, **H14** is accepted, while **H15** is rejected.

The OLS analysis results of the hypothesis testing for hypotheses **H10** to **H15** are summarized in Table 10.

**Table 10.** Hypothesis testing from **H10** to **H15**.

| Hypothesis | Coefficient | Std Err | t-Value | *p*-Value | Result |
|:---:|:---:|:---:|:---:|:---:|:---:|
| **H10** | 0.0536 | 0.008 | 6.505 | 0.017071 | Reject the null hypothesis |
| **H11** | 0.0036 | 0.016 | 0.230 | 0.081823 | Failed to reject the null hypothesis |
| **H12** | 0.2152 | 0.041 | 5.221 | 0.002171 | Reject the null hypothesis |
| **H13** | 0.0362 | 0.061 | 0.593 | 0.0653 | Failed to reject the null hypothesis |
| **H14** | 0.2480 | 0.014 | 17.553 | 0.01912 | Reject the null hypothesis |
| **H15** | 0.0050 | 0.041 | 0.121 | 0.09034 | Failed to reject the null hypothesis |

In summary, the OLS was analyzed to examine the influence of tourist characteristics on the impact of MGC and UGC when using social media for planning travel to Saudi Arabia. The findings revealed that tourist characteristics impact MGC but not UGC in terms of their intention to use social media for travel planning.

5.5.3. The Influence of the Visit Characteristics

To investigate how visit characteristics affect the tourists' intentions to use social media to plan travel to Saudi Arabia, the hypotheses were tested using OLS. **H16** and **H17** examine how MGC and UGC affect tourists' intentions to use social media for planning travel to Saudi Arabia, differing depending on whether tourists have visited the country before. Based on the OLS results of "**H16**: *The effect of MGC on tourists' intentions to use social media for planning travel to Saudi Arabia will differ based on whether they have made previous visits*", the coefficient is 0.4007, with a standard error of 0.058, a t-value of 6.857, and a *p*-value of 0.01833, which means that there is a significant difference between the effect of MGC on tourists' intentions to use social media for planning travel to Saudi Arabia based on whether they have made previous visits. MGC can help to build trust and confidence among these tourists, making them more likely to use social media for travel planning. Meanwhile, the OLS results of "**H17**: *The effect of UGC on tourists' intentions to use social media for planning travel to Saudi Arabia will differ based on whether they have made previous visits*" show that the coefficient is −0.2472, with a standard error of 0.108, a t-value of −2.293, and *p*-value of 0.07822, which means there is no significant difference between the effect of MGC on tourists' intentions to use social media for planning travel to Saudi Arabia based on whether they have made previous visits. UGC can create confidence among all tourists when using social media for travel planning. Therefore, **H16** is accepted, while **H17** is rejected.

**H18** and **H19** examine how MGC and UGC affect tourists' intentions to use social media for planning travel to Saudi Arabia, differing depending on the season that tourists prefer. The OLS results of "**H18**: *The effect of MGC on tourists' intentions to use social media for planning travel to Saudi Arabia will differ by season*" show that the coefficient is 0.0672, with a standard error of 0.065, a t-value of 13.822, and a *p*-value of 0.00122, which means there is a significant difference in the effect of MGC on tourists' intentions to use social media for planning travel to Saudi Arabia based on the season that they prefer. The OLS results of "**H19**: *The effect of UGC on tourists' intentions to use social media for planning travel to Saudi Arabia will differ by season*" show that the coefficient is 0.3676, with a standard error of 0.027, a t-value of 1.031, and a *p*-value of 0.06303, which means there is no significant difference in the effect of UGC on tourists' intentions to use social media for planning travel to Saudi Arabia based on the season that they prefer. Therefore, **H18** is accepted, while **H19** is rejected.

**H20** and **H21** examine how MGC and UGC affect tourists' intentions to use social media for planning travel to Saudi Arabia, differing depending on the length of stay. Based on the OLS results of "**H20**: *The effect of MGC on tourists' intentions to use social media for planning travel to Saudi Arabia will differ based on the length of stay*", the coefficient is 0.2652, with a standard error of 0.030, a t-value of 8.868, and a *p*-value of 0.0458, indicating that the length of stay influences the impact of MGC on tourists' intentions to use social media for planning travel in Saudi Arabia. The survey results have shown that most respondents

(46.88%) favored a seven-day stay; this indicates that if tourists plan a brief stay in Saudi Arabia, they may rely on utilizing MGC to obtain comprehensive information about Saudi Arabia. The OLS results of "**H21**: *The effect of UGC on tourists' intentions to use social media for planning travel to Saudi Arabia will differ based on the length of stay*" show a coefficient of 0.0897, with a standard error of 0.064, a t-value of 1.404, and a *p*-value of 0.08161, indicating that the length of stay does not affect the impact of UGC on tourists' intentions to use social media for planning travel in Saudi Arabia. Therefore, **H20** is accepted, while **H21** is rejected.

Further, **H22** and **H23** examine how MGC and UGC affect tourists' intentions to use social media for planning travel to Saudi Arabia, differing depending on the type of accommodation. The OLS results of "**H22**: *The effect of MGC on tourists' intentions to use social media for planning travel to Saudi Arabia will differ based on the type of accommodation*" show a coefficient of 0.2648, with a standard error of 0.019, a t-value of 13.802, and a *p*-value of 0.00138, indicating that the type of accommodation influences the impact of MGC on tourists' intentions to use social media for planning travel to Saudi Arabia. The reliance of tourists on MGC is different based on the accommodation type. According to the survey analysis, tourists looking for a motel or economical hotel will rely on MGC more than other tourists. The OLS results of "**H23**: *The effect of UGC on tourists' intentions to use social media for planning travel to Saudi Arabia will differ based on the type of accommodation*" show that the coefficient is 0.0363, with a standard error of 0.047, a t-value of 0.775, and a *p*-value of 0.094391, indicating that the type of accommodation does not affect the impact of UGC on tourists' intentions to use social media for planning travel in Saudi Arabia. Therefore, **H22** is accepted, while **H23** is rejected.

In summary, the results of the OLS analysis showed that visit characteristics affect MGC but not UGC when it comes to tourists' intentions to use social media for travel planning. Table 11 summarizes the findings of the OLS analysis and hypothesis testing.

**Table 11.** Hypothesis testing from **H16** to **H23**.

| Hypothesis | Coefficient | Std Err | t-Value | *p*-Value | Result |
|---|---|---|---|---|---|
| **H16** | 0.4007 | 0.058 | 6.857 | 0.01833 | Reject the null hypothesis |
| **H17** | −0.2472 | 0.108 | −2.293 | 0.07822 | Failed to reject the null hypothesis |
| **H18** | 0.0672 | 0.065 | 13.822 | 0.00122 | Reject the null hypothesis |
| **H19** | 0.3676 | 0.027 | 1.031 | 0.06303 | Failed to reject the null hypothesis |
| **H20** | 0.2652 | 0.030 | 8.868 | 0.0458 | Reject the null hypothesis |
| **H21** | 0.0897 | 0.064 | 1.404 | 0.08161 | Failed to reject the null hypothesis |
| **H22** | 0.2648 | 0.019 | 13.802 | 0.00138 | Reject the null hypothesis |
| **H23** | 0.0363 | 0.047 | 0.775 | 0.094391 | Failed to reject the null hypothesis |

*5.6. Tourist-Based ML Classification Model*

The tourist-based ML classification model was trained using a combination of selected classification algorithms. The MGC class is labelled 0, and the UGC class is labelled 1. The performance of each model was assessed by calculating its accuracy, precision, recall, f1-score, and support for each class label.

5.6.1. Linear Classification Models

Linear classification models, including logistic regression, LDA, and LinearSVC, were trained. The logistic regression classification model achieved an accuracy score of 0.9593, indicating a correct prediction of class labels for (95.93) of test instances. The model showed strong results, with high precision and recall scores for both classes, with slightly better scores for class UGC (0.98 and 0.95) than for class MGC (0.93 and 0.97). The f1-score improved for both classes, with class UGC (0.97) showing slightly better results than class MGC (0.95). The support indicated more instances in class UGC (108) than in class MGC (64), suggesting that the model performed well in distinguishing between the two classes.

The LDA classification model achieved an accuracy score of 0.9070, indicating a correct prediction of class labels for 90.70% of test instances. The model demonstrated high precision and recall scores for both classes, with better scores for class UGC (0.94) and (0.91) than for class MGC (0.85 and 0.91). The f1-score also improved for both classes, with class UGC (0.92) showing slightly better results than class MGC (0.88). The support indicated more instances in class UGC (108) than in class MGC (64), suggesting that the model performed well in distinguishing between the two classes.

The LinearSVC classification model achieved an accuracy score of 0.9767, indicating that the model accurately predicted the class labels for 97.67% of the test cases. For class UGC, the model achieved a perfect precision score of 1.00, meaning that the model made no false positive errors for this class. However, the model missed 4% of the actual positives for the UGC class, as indicated by the recall score of 0.96. The f1-score for class UGC was 0.98, indicating a good balance between precision and recall. There were 108 instances of the UGC class in the test set. For class MGC, the model achieved a precision score of 0.94, indicating that the model made 6% false positive errors for the MGC class. However, the model did not miss any actual positives for the MGC class, as indicated by the recall score of 1.00. The f1-score for class MGC was 0.97, indicating a good balance between precision and recall. There were 64 instances of the MGC class in the test set. Overall, the model demonstrated high accuracy and performance for both classes.

### 5.6.2. Tree-Based Classification Models

The tree-based classification models trained included the decision tree classifier, extra tree classifier, and random forest classifier. The decision tree classifier model achieved an accuracy of 0.9069, indicating its ability to predict target variable values for 90.69% of the test instances. The classification report reveals that class MGC had a precision of 0.84 and class UGC had a precision of 0.95, which signifies that the model can accurately identify the presence or absence of a specific attribute in the data. In addition, the model had a recall score of 0.92 for class MGC and 0.90 for class UGC, indicating its ability to detect all instances of the target class accurately. The f1-score for class MGC was 0.88 and for class UGC was 0.92, indicating a good balance between precision and recall. The support value indicates that 64 instances belonged to class MGC and 108 to class UGC, suggesting that the model performed well in distinguishing between the two classes.

The extra tree classifier model performed well, with an accuracy of 0.8837, correctly predicting 88.4% of the test cases. The classification report shows that the model had a precision for MGC of 0.83 and UGC 0.92, indicating that it could accurately distinguish between positive and negative instances. The recall for MGC (0.86) and UGC (0.90) indicates that it could accurately distinguish between positive and negative instances. The f1-score, which measures the balance between precision and recall, was MGC at 0.85 and UGC at 0.91. The support shows 64 instances of MGC and 108 instances of UGC in the test set.

The random forest classifier model achieved an accuracy of 0.9476, correctly predicting 94.77% of the test cases. The model had a high precision for both classes—0.91 for MGC and 0.97 for UGC—meaning that it had a low rate of false positives. The model also had a high recall for both classes—0.95 for MGC and 0.94 for UGC—meaning that it had a low rate of false negatives. The f1-score measures the balance between precision and recall, and the model had a high f1-score for both classes: 0.93 for MGC and 0.96 for UGC. The support shows 64 instances of MGC and 108 instances of UGC in the test set.

### 5.6.3. MLP and KNN Models

The MLP and KNN models were trained. The MLP classifier model achieved an accuracy of 0.8546, correctly predicting 85.5% of the test cases. The classification report shows that the model had a moderate precision for both classes—0.77 for MGC and 0.91 for UGC—meaning that it had a moderate rate of false positives. The model also had a recall 0.86 for MGC and 0.85 for UGC, meaning that it had a low rate of false negatives.

The f1-scores for MGC and UGC were 0.81 and 0.88, respectively. The support shows 64 instances of MGC and 108 instances of UGC in the test set.

The KNN classifier model achieved an accuracy of 0.8662, meaning that it correctly predicted 86.63% of the test cases. The classification report shows that the model had a moderate precision for both classes—0.84 for MGC and 0.88 for UGC—meaning that it had a moderate rate of false positives. The model also had a high recall for both classes—0.80 for MGC and 0.91 for UGC—meaning it had a low rate of false negatives. The f1-scores for MGC and UGC were 0.82 and 0.89, respectively. The support shows 64 MGC instances and 108 instances of UGC in the test set.

Based on the performance results displayed in Table 12, it was observed that the LinearSVC, random forest classifier, and logistic regression models exhibited superior performance. Consequently, these models were selected for further evaluation in the next phase to determine the best ML classification algorithm for the tourist-based classification model.

**Table 12.** Performance results of the models.

| Model | Accuracy | Precision | Recall | F1-Score |
| --- | --- | --- | --- | --- |
| Logistic Regression | 0.9593 | 0.98 | 0.95 | 0.97 |
| LDA | 0.9070 | 0.94 | 0.91 | 0.92 |
| LinearSVC | 0.9767 | 1.00 | 0.96 | 0.98 |
| Decision tree | 0.9069 | 0.84 | 0.92 | 0.88 |
| Extra tree | 0.8837 | 0.83 | 0.86 | 0.85 |
| Random forest | 0.9476 | 0.91 | 0.95 | 0.93 |
| MLP | 0.8546 | 0.77 | 0.86 | 0.81 |
| KNN | 0.8662 | 0.84 | 0.80 | 0.82 |

*5.7. Tourist-Based ML Model Evaluation*

This study aimed to determine the most effective ML model for classifying tourist data. Three models, namely logistic regression, random forest classifier, and LinearSVC, were evaluated using the K-Fold Cross Validation (KF-CV) approach to ensure accuracy and reliability. After thoroughly analyzing the data, the results showed that the random forest classifier and logistic regression models achieved high accuracy scores of 0.95, even after applying KF-CV. However, the LinearSVC model stood out with an exceptional accuracy score of 0.99, indicating its superiority in avoiding overfitting compared to the other models. The results of the accuracy calculated after KF-CV was applied are shown in Table 13.

**Table 13.** The results of the accuracy of the models.

| Model | Accuracy | Evaluated Accuracy |
| --- | --- | --- |
| Logistic Regression | 0.95 | 0.95 |
| Random Forest | 0.94 | 0.95 |
| LinearSVC | 0.97 | 0.99 |

LinearSVC is a highly versatile and practical ML model that can be used for classification and regression tasks. One of its key strengths is using a linear kernel, which calculates the dot product of two feature vectors, allowing for the efficient processing of large datasets and ensuring that the model can make accurate predictions, even when working with complex and nuanced data [52]. In particular, the LinearSVC model is particularly well-suited for working with linearly separable data. When a straight line can separate data, the model can achieve perfect accuracy, making it an ideal choice for a wide range of classification tasks. Additionally, the LinearSVC model is rapid and efficient regarding training and implementation, making it an excellent choice for working with large datasets [52]. Another key advantage of the LinearSVC model is its interpretability. Unlike some ML models, the LinearSVC model is developed to be easy to understand and explain, making it a valuable

tool for researchers, analysts, and other professionals who need to be able to interpret and make sense of their results.

Overall, the LinearSVC model is a highly accurate and reliable choice for correctly classifying tourist data, given its high accuracy, resistance to overfitting, and interoperability. Its use can lead to the development of more effective and efficient Saudi tourism-related content and digital marketing strategies on social media platforms, meaning that this research significantly contributes to ML, tourism, and marketing.

## 6. Discussion

The significance of this study is to explain how tourists use social media platforms to plan their travels to Saudi Arabia and what variables impact their intentions and decisions, as these still need to be investigated according to the literature and past studies. This goal is achieved with the results presented in Section 5. First, the fourth chapter presents the results of the analysis of the variables that influence the intentions and decisions of tourists, including PU, PEOU, SAT, MGC, and UGC, as independent variables influencing their intentions to use social media in planning travel to Saudi Arabia. Subsequently, this study examines the effects of tourists' characteristics and visit characteristics on the use of UGC and MGC on social media platforms to plan travel to Saudi Arabia. Second, the fourth chapter shows the results of applying the selected algorithms to build and evaluate the tourist-based ML classification model.

### 6.1. Tourist-Based Theoretical Framework

This study conducted a thorough survey that involved forty-six questions, carefully analyzing and assessing the data to gain valuable insights. The findings of the hypothesis testing of the tourist-based theoretical framework addressed the first three research questions. The first research question was "What variables impact tourists' intentions to use social media platforms for planning travel to Saudi Arabia?" Five hypotheses ($H_1$–$H_5$) were developed to answer the first research question by investigating the impact of PU, PEOU, SAT, MGC, and UGC on tourists' intentions to use MGC and UGC on social media platforms for planning travel to Saudi Arabia.

The results of the OLS regression analysis showed that PU, PEOU, SAT, MGC, and UGC all significantly positively impact tourists' intentions to use social media platforms for planning travel to Saudi Arabia. This is in line with earlier findings that demonstrate the impact of these variables on tourists' intentions to use social media platforms for travel planning [35,36,53]. Therefore, Saudi tourism SMEs should consider the relationship between these variables and tourists' intentions to use social media platforms for planning travel to Saudi Arabia in developing their digital marketing strategy on social media. By concentrating on these variables and understanding how they influence tourists' behavior, SMEs can develop an effective and targeted marketing strategy to attract more tourists to Saudi Arabia. For example, SMEs can leverage the PU of social media platforms by providing useful information about tourist destinations in Saudi Arabia. They can also improve the PEOU of social media platforms by choosing platforms that are more user-friendly and accessible. In addition, SMEs can enhance tourists' satisfaction by providing high-quality content that meets their needs and expectations. Finally, SMEs can leverage MGC and UGC to create more engaging and interactive social media campaigns to attract tourists.

The second research question was "What is the impact of the tourists' characteristics on their intentions to use MGC and UGC on social media platforms for planning travel to Saudi Arabia?" To answer the second research question, ten hypotheses ($H_6$–$H_{15}$) were developed to investigate the impact of tourists' characteristics on tourists' intentions to use MGC and UGC on social media platforms for planning travel to Saudi Arabia. The findings of the OLS regression analysis revealed that the characteristics of tourists influenced their intentions to use MGC on social media platforms for planning travel to Saudi Arabia. This study's findings align with prior research that has established the link between the characteristics of tourists and their intentions to use MGC on social media platforms for travel

planning [37,54,55]. However, the characteristics of the tourists did not influence their intentions to use UGC on social media platforms for planning travel to Saudi Arabia. These findings contradict the hypothesis that the characteristics of tourists influence their intentions to use UGC on social media platforms for travel planning [40,56,57]. Therefore, Saudi tourism SMEs should consider that the characteristics of tourists are necessary variables that will determine the preferred content type to use in developing their digital marketing strategy on social media. If SMEs mainly target tourists based on their characteristics, MGC is preferred over UGC. Saudi tourism SMEs might use MGC to target a particular category of tourists, such as luxury travelers, teenage travelers, females, or certain nationalities of travelers. For example, Saudi tour-guide SMEs target tourists from Arab nationalities, since they do not have tour guides who speak other languages. In this case, Saudi tour-guide SMEs should use MGC in Arabic to publish their services on social media through the official tour-guide SMEs' accounts.

As a result, this study suggests that MGC might be an effective digital marketing strategy for Saudi tourism SMEs that understand their target market and identify the target audience that is likely to be interested in their products or services. Otherwise, UGC might be an effective digital marketing strategy for Saudi tourism SMEs that do not target a particular category of tourists.

The third research question was "What is the impact of visit characteristics on tourists' intentions to use MGC and UGC on social media platforms for planning travel to Saudi Arabia?" Eight hypotheses ($H_{16}$–$H_{23}$) were developed to answer the third research question to investigate the impact of the visit characteristics on tourists' intentions to use MGC and UGC on social media platforms for planning travel to Saudi Arabia. The findings of the OLS regression analysis revealed that the visit characteristics influenced their intentions to use MGC on social media platforms for planning travel to Saudi Arabia. However, the visit characteristics did not influence tourists' intentions to use UGC on social media platforms for planning travel to Saudi Arabia. This study's findings corroborate previous research on the relationship between visit characteristics and tourists' intentions to use MGC on social media platforms for travel planning. However, these findings contradict the findings of other studies that suggest that visit characteristics influence tourists' intentions to use UGC on social media platforms for travel planning [38,43,58,59]. Therefore, Saudi tourism SMEs should consider the visit characteristics necessary to determine the preferred content type for developing their digital marketing strategy on social media. If SMEs mainly target tourists based on the visit characteristics of tourists, MGC is preferred over UGC. Saudi tourism SMEs might use MGC to target a particular category of tourists based on the visit characteristics of tourists, such as travelers with previous visits, travelers who prefer economic hotels to luxury hotels, travelers who prefer to stay more than two weeks, or travelers who prefer to stay less than one week. For example, Saudi tourism SMEs target travelers who prefer to stay less than one week since they have limited activities. In this case, Saudi tourism SMEs should use MGC about their activities with the duration time of the trip to publish on social media through the official tour-guide SMEs' accounts. As a result, this study suggests that MGC might be an effective digital marketing strategy for Saudi tourism SMEs that target tourists based on their visit characteristics. Otherwise, UGC might be an effective digital marketing strategy for Saudi tourism SMEs that do not target a particular category of tourists based on their visit characteristics.

### 6.2. Tourist-Based ML Classification Model

The tourist-based ML classification model was developed and evaluated to answer the fourth research question: "How can ML be used to improve the digital marketing strategies of Saudi tourism SMEs on social media platforms?". This study found that ML can be particularly useful for Saudi tourism SMEs that desire to develop their marketing strategies to target tourists. According to the tourist-based ML classification model results, the LinearSVC classification algorithm is an excellent choice to segment tourists based on whether they prefer MGC or UGC. The LinearSVC algorithm has a high accuracy score and

consistent performance, which means that the LinearSVC algorithm is likely to accurately predict whether a tourist prefers MGC or UGC, even if the dataset is small.

In light of the above, the tourist-based ML classification model can classify tourists depending on their preferences for MGC or UGC. Tourist segmentation can help Saudi tourism SMEs reach out to different types of tourists depending on their demographics, interests, and behavior. This segmentation can assist Saudi tourist SMEs in developing their digital marketing strategy with a focus on social media. This data may be utilized to develop better-focused marketing campaigns on social media that are more likely to resonate with each tourist's demographic and preference for visit characteristics.

*6.3. Developing Digital Marketing Strategies*

Digital marketing generates and distributes great content to attract tourists [60]. It creates trust and motivates tourists to seek help from Saudi tourist SMEs. Content on social media platforms could help to answer relevant tourists' questions or explore topics of interest. The fifth research question was "What recommendations can help to improve the digital marketing strategies of Saudi tourism SMEs on social media platforms?" Based on the results of this study and theoretical insights, the following recommendations can be suggested regarding empirical data and theoretical insights for Saudi tourism SMEs to improve their digital marketing strategy on social media:

- Provide relevant and valuable information about tourist attractions in Saudi Arabia to increase the perceived usefulness of social media platforms;
- Select social media platforms that are more compatible with tourists' preferences and devices to enhance the perception of ease of use;
- Deliver high-quality content that meets tourists' needs and expectations to improve their satisfaction;
- Create engaging and interactive social media campaigns involving tourists cocreating and sharing content to leverage MGC;
- Encourage and facilitate tourists to create and share their content about their travel experiences in Saudi Arabia to leverage UGC;
- MGC can be used as a content-marketing tool to target specific segments of tourists based on their characteristics and visit characteristics. Moreover, the results suggest that Saudi tourism SMEs can leverage the advantages of MGC as a content type that can be controlled and customized by them to suit their brand image and message, as well as their business objectives and strategies. For instance, they can use MGC to showcase cultural and historical attractions that reflect the heritage and identity of Saudi Arabia, provide relevant and valuable information about travel planning and preparation in Saudi Arabia and address tourists' questions or concerns, and feature the various activities and attractions that can be enjoyed in Saudi Arabia within a certain length of stay. These measures can showcase the diversity and richness of the Saudi tourism sector;
- Saudi tourism SMEs should not rely solely on MGC as a content type that may not appeal to all tourists or reflect their experiences. Instead, SMEs should use UGC as complementary content to enhance the credibility and authenticity of their social media presence;
- UGC suits Saudi tourism SMEs needing more knowledge about their target market and audience. The results of this study indicated that UGC is not influenced by the characteristics of tourists, which implies that it can have a wider reach among all tourists, regardless of their demographics. Consequently, tourists interested in the service or product can contact Saudi tourism SMEs. However, the results suggest that there may be better long-term strategies than relying on UGC, as this type of content requires targeted marketing. An effective marketing strategy should target potential customers, which can yield more returns than non-targeting. Therefore, this study suggests that companies can use UGC but, at the same time, they should collect and

analyze data about their customers and create databases to segment and understand them to facilitate targeting and access to them.

## 7. Conclusions

This study investigated the significance of employing ML to understand tourists' intent to use social media for planning travel to Saudi Arabia. A tourist-based theoretical framework was proposed to identify tourists' intentions to use social media for planning travel to Saudi Arabia. In addition, the tourist-based ML classification model was developed to assist Saudi tourist SMEs in developing effective digital marketing strategies for social media platforms. This study adopted the DSR approach and surveyed 573 tourists interested in visiting Saudi Arabia.

The tourist-based theoretical framework was supported by the findings of this study, which showed that PU, PEOU, SAT, MGC, and UGC significantly impact tourists' intentions to use social media platforms for planning travel to Saudi Arabia. Also, the findings revealed that tourists' characteristics influenced their intentions to use MGC but not UGC on social media platforms for planning travel to Saudi Arabia. In addition, the visit characteristics influenced the intentions of tourists to use MGC but not UGC on social media platforms for planning travel to Saudi Arabia.

The tourist-based ML classification model was developed using the LinearSVC algorithm due to its high accuracy score and reliable performance. The model was evaluated using the KF-CV technique, which showed that the model had an accuracy of 99%. The results indicate that the tourist-based ML classification model is a reliable and effective tool for classifying tourists depending on their preferences for MGC or UGC.

The findings of this study could have several implications for Saudi tourism SMEs to develop an effective digital marketing strategy. First, the findings suggest that SMEs should consider using social media platforms to market their businesses. Second, SMEs should invest in understanding the variables that impact their target audience. Third, SMEs could utilize ML to understand tourists' intentions to use social media in travel planning and classify them.

## 8. Limitations and Future Research

This study has several limitations. First, the sample size was relatively small, and a larger sample would be more generalizable to a larger population. Second, the study focused on tourists' intentions to use social media for travel planning, and the sample may need to be more generalizable to other aspects of tourism marketing. Third, the tourist-based theoretical framework was developed based on specific variables, and the results may need to be tested with more variables and characteristics that might impact tourists' intentions. Fourth, the study used closed-ended questions, which limited the depth of the respondents' answers. It would have been better to add some open-ended questions to allow respondents to add their own ideas and perspectives. Fifth, this study investigated Saudi tourism SMEs in general, and the results may need to be more narrowly specified in order to focus on dedicated sectors.

The results of this study offer important insights into the factors that influence tourists' intentions to use social media for travel planning. However, it is important to consider the limitations of this study when interpreting the results. Future research that addresses these limitations could provide further insights into this topic. One limitation of this study is the sample size. A larger sample size would increase the generalizability of the findings. Additionally, this study was conducted in a single context. Future research should be conducted in other contexts to test the generalizability of the results. Also, this study focused on tourists' intentions to use social media for travel planning. Future research should focus on other aspects of tourism marketing, such as tourists' actual use of social media for travel planning and the impact of social media on tourists' travel decisions. In addition to the above, future researchers can also build studies based on the results of the proposed model and conduct marketing studies on classified groups, such as the

study of preferred tourists for MGC and those preferred for UGC. One such approach could be proposing ML models that depend on aggregation for each category to target them more in marketing or search for common characteristics between them to develop marketing strategies that combine MGC and UGC and are based on common factors among tourists in both groups. The findings also suggest that future researchers should investigate how to develop the tourist-based ML classification model and build similar models using artificial intelligence chatbots, such as ChatGPT and Bard. We also suggest studying the development of tourism marketing in Saudi Arabia by using artificial intelligence in creating marketing content and studying the impact of this content on the intentions of tourists and their decisions when planning to travel to Saudi Arabia.

**Author Contributions:** Conceptualization: R.A.A. and B.F.; methodology: R.A.A. and B.F.; investigation: R.A.A. and B.F.; data curation: R.A.A. and B.F.; statistical analysis: R.A.A. and B.F.; writing—original draft preparation: R.A.A. and B.F.; writing—review and editing: R.A.A. and B.F.; supervision: B.F. All authors have read and agreed to the published version of the manuscript.

**Funding:** This research received no external funding.

**Institutional Review Board Statement:** Not applicable.

**Informed Consent Statement:** Not applicable.

**Data Availability Statement:** Data sharing in not applicable to this article.

**Conflicts of Interest:** The authors declare no conflict of interest.

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
