# Peer review of "A Tourist-Based Framework for Developing Digital Marketing for Small and Medium-Sized Enterprises in the Tourism Sector in Saudi Arabia"

_data, 2023_

Round 1

Reviewer 1 Report

Comments and Suggestions for Authors

Dear authors, I liked your work, it is interesting, the way of writing is correct and I think the readers will love the work. Although I see some shortcomings in your work. Below are my comments to improve the article.

- The article has many meaningless sub-chapters, it is not a final degree or master's thesis, it is a research paper and it has to be very clear and concrete. In particular, the introduction part has to be a clear part where you would have the motivation and justification for choosing the topic, objectives of the work, methods, results and conclusions and implications. There is no need to have subchapters within the introduction, it should be a complete text and maximum three pages.

- The same thing happens in the literature review section, I would put-- Literature review and directly 2.1. Digital Marketing, no need for introductions and three line subchapters.

- You should separate the hypotheses section from the review of literature, sincerely it seems to me that the 23 hypotheses that you are presenting are too much, you should summarise and justify the hypotheses better.

- Please provide the source for the figures

- The discussion of the results involves comparison with the literature analysed above, you just explain your result.

- I also find the data analysis part too much, it is really the description of the sample pointing out the accurance of the model but at no point do we see a regression analysis (or any other type of analysis) where we can observe the reality of the variables and the model proposed.

- Once again I think that the paper is in the format of a final thesis, the authors have a big problem with the length of the paper, a scientific article should be about 8 thousand words.

- The following research paper could serve you justifing the election of the topic and also can serve asthe examples how to write and modificate your article.

Debasa, F., Gelashvili, V., Martínez-Navalón, J. G., & Saura, J. R. (2023). Do stress and anxiety influence users' intention to make restaurant reservations through mobile apps?. European Research on Management and Business Economics, 29(1), 100205.

Wish you good luck

Comments on the Quality of English Language

English language quality is good. Proofread is required

Author Response

Reviewer#1, Concern # 1: - The article has many meaningless sub-chapters, it is not a final degree or master's thesis, it is a research paper and it has to be very clear and concrete. In particular, the introduction part has to be a clear part where you would have the motivation and justification for choosing the topic, objectives of the work, methods, results and conclusions and implications. There is no need to have subchapters within the introduction, it should be a complete text and maximum three pages.

Author response:  We do agree with the reviewer. Thank you.

Author action: We managed to summarize the introduction and remove the subchapters.

Reviewer#1, Concern # 2: - The same thing happens in the literature review section, I would put-- Literature review and directly 2.1. Digital Marketing, no need for introductions and three line subchapters.

Author response:  We do agree with the reviewer. Thank you.

Author action: We managed to remove the introductions and three-line subchapters in the literature review section.

Reviewer#1, Concern # 3:  You should separate the hypotheses section from the review of literature, sincerely it seems to me that the 23 hypotheses that you are presenting are too much, you should summarise and justify the hypotheses better.

Author response:  We do agree with the reviewer, but the hypotheses highlight the same aims of the study and the developed model in concern two, and the results of the model present that. Thank you.

Author action: We managed to enhance the justification and summarize the hypotheses, and we separated the hypotheses in a new section.

Reviewer#1, Concern # 4:  Please provide the source for the figures.

Author response:  Thank you for your comment, but all figures have resulted from this study; except Figure 4.1. As well, Figure 4.1 was deleted based on Reviewer 4's comments and recommendations.

Author action: No action was taken.

Reviewer#1, Concern # 5:  The discussion of the results involves comparison with the literature analysed above, you just explain your result.

Author response:  Thank you for the comment. I agree that it is important to compare one's results to the existing literature in the discussion section of a paper. This helps to contextualize the findings and identify any new or unexpected insights. I believe that these comparisons make the discussion of the results section stronger and more informative.

Author action: No action was taken.

Reviewer#1, Concern # 6:  I also find the data analysis part too much, it is really the description of the sample pointing out the accurance of the model but at no point do we see a regression analysis (or any other type of analysis) where we can observe the reality of the variables and the model proposed.

Author response:  Thank you for the comment. In addition to describing the sample, we used inferential statistics including Cronbach's alpha to assess the internal consistency of the constructs. Also, we used Correlation analysis is a valuable statistical tool for determining the strength and direction of the linear relationship between two variables, as well as evaluating their association.

Author action: No action was taken.

Reviewer#1, Concern # 7:  Once again I think that the paper is in the format of a final thesis, the authors have a big problem with the length of the paper, a scientific article should be about 8 thousand words.

Author response:  With the total respect to the reviewer, the paper contains a lot of important findings that we could not eliminate. Also, the paper fits within the accepted range of the journal.  We hope this would be accepted.  

Author action: No action was taken.

Reviewer#1, Concern # 8:  - The following research paper could serve you justifing the election of the topic and also can serve asthe examples how to write and modificate your article.

Author response:  Thank you for the comment. We read the paper mentioned in the comment, and we will use it in our upcoming paper.

Author action: No action was taken

Reviewer 2 Report

Comments and Suggestions for Authors

The article tackles a very interesting and topical issue. Carefully described the research process in sequential steps. An elaborate research model was used. Precise formulation of research objectives, and problems. Please complete the information on the selection of the research sample. How was it purposive, random, convenient etc. Very interesting article with useful research. The authors showed great care in analysing the results. There was also a valuable analysis of the literature on the subject. The tourism market in the study area was presented.

The whole is methodologically correct.

The limitations and implications of the research are worth further elaboration. The article should be published as valuable research material.

Author Response

Reviewer#2, Concern # 1: The article tackles a very interesting and topical issue. Carefully described the research process in sequential steps. An elaborate research model was used. Precise formulation of research objectives, and problems. Please complete the information on the selection of the research sample. How was it purposive, random, convenient etc. Very interesting article with useful research. The authors showed great care in analysing the results. There was also a valuable analysis of the literature on the subject. The tourism market in the study area was presented.

The whole is methodologically correct.

Author response:  Thank you very much for the comment.

Author action: We have rewritten and enhanced the paragraph regarding the sample of the paper by justifying and adding more information on the selection of the research sample. How was it purposive, random, convenient.

Reviewer#2, Concern # 2: The limitations and implications of the research are worth further elaboration. The article should be published as valuable research material.

 Author response:  Thank you very much for the comment, and sorry for that.

Author action: We have rewritten and enhanced the limitations and implications of the research.

Reviewer 3 Report

Comments and Suggestions for Authors

This is an excellent paper outside of a minor typo or two. My only concern is that the paper seems to be taken from a master's thesis and reads like one. The author uses the term "chapter" to refer to the sections of the paper. Each section ends with a statement summarizing what was done in the chapter and what the next chapter would entail. Then, the next section opens up with the author describing what was going to happen in this chapter. I would recommend that the wording of the beginning and ending of each section be changed so that the paper reads as a paper and not a thesis. Other than that, the author is to be commended for this excellent piece of research.

Author Response

Reviewer#3, Concern # 1: This is an excellent paper outside of a minor typo or two. My only concern is that the paper seems to be taken from a master's thesis and reads like one. The author uses the term "chapter" to refer to the sections of the paper. Each section ends with a statement summarizing what was done in the chapter and what the next chapter would entail. Then, the next section opens up with the author describing what was going to happen in this chapter. I would recommend that the wording of the beginning and ending of each section be changed so that the paper reads as a paper and not a thesis. Other than that, the author is to be commended for this excellent piece of research.

Author response:  Thank you very much for the comment.

Author action: We have rewritten and enhanced the paper and converted the term "chapter" to the term “section” in the paper. As well, we removed the beginning and ending of each section.

Reviewer 4 Report

Comments and Suggestions for Authors

Dear authors,

This is an interesting manuscript on the development of digital marketing for SMEs in the tourism sector.

The manuscript is comprehensively structured, and all sections are meticulously explored.

In Section 1. Introduction, the sentence "Nowadays, social media platforms play a critical role in researching destinations, booking accommodations, and sharing experiences" is cited as [3]. However, the reference listed as [3] in the References section is "V. Yılancı and M. Kirca, 'Testing the relationship between employment and tourism: a fresh evidence from the ardl bounds test with sharp and smooth breaks,' Journal of Hospitality and Tourism Insights, 2023." This paper does not address the issue of social networks, so the citation is not correct. It is recommended that the authors review all citations throughout the manuscript to ensure their accuracy.

Figure 3.1 does not add value to the discussion. It is suggested that this figure be deleted.

The questionnaire in Appendix A is well-structured, and a pilot study on it was conducted with a random sample of 10 tourists. The authors are asked to clarify whether the questionnaire's face and content validity were assessed. If so, by whom? This is an important issue that must be addressed in the manuscript.

In Section 4. Data Analysis and Results, the following two recommendations are made: (i) Delete Figure 4.8, because the data are already presented in Table 4.3. (ii) In Tables 4.7, 4.8, and 4.9, use the representation 0.0894884 instead of 8.94884x10(-2) in the p-value column.

Finally, in Section 5.1.3, Developing Digital Marketing Strategies, the authors are asked to clarify whether the listed recommendations resulted from the research conducted.

Well done.

Author Response

Reviewer#4, Concern # 1: In Section 1. Introduction, the sentence "Nowadays, social media platforms play a critical role in researching destinations, booking accommodations, and sharing experiences" is cited as [3]. However, the reference listed as [3] in the References section is "V. Yılancı and M. Kirca, 'Testing the relationship between employment and tourism: a fresh evidence from the ardl bounds test with sharp and smooth breaks,' Journal of Hospitality and Tourism Insights, 2023." This paper does not address the issue of social networks, so the citation is not correct. It is recommended that the authors review all citations throughout the manuscript to ensure their accuracy.

 Author response:  Thank you very much for the comment, and yes this was mistakenly written.

Author action: We corrected the mistake, and we reviewed all citations throughout the manuscript.

Reviewer#4, Concern # 2: Figure 3.1 does not add value to the discussion. It is suggested that this figure be deleted.

 Author response:  Thank you very much for the comment, and sorry for that.

Author action: We deleted the Figure 3.1.

Reviewer#4, Concern # 3: The questionnaire in Appendix A is well-structured, and a pilot study on it was conducted with a random sample of 10 tourists. The authors are asked to clarify whether the questionnaire's face and content validity were assessed. If so, by whom? This is an important issue that must be addressed in the manuscript.

 Author response:  Thank you very much for the comment, and sorry for that.

Author action: We have rewritten and enhanced the paragraph of the pilot study and added more details about the random sample that was selected.

Reviewer#4, Concern # 4: In Section 4. Data Analysis and Results, the following two recommendations are made: (i) Delete Figure 4.8, because the data are already presented in Table 4.3. (ii) In Tables 4.7, 4.8, and 4.9, use the representation 0.0894884 instead of 8.94884x10(-2) in the p-value column.

 Author response:  Yes, we do agree with the valuable comment from the reviewer.

Author action: We deleted Figure 4.8. As well, we rewrite the p-value column in Tables 4.7, 4.8, and 4.9, using the representation 0.0894884 instead of 8.94884x10(-2).

Reviewer#4, Concern # 5: Finally, in Section 5.1.3, Developing Digital Marketing Strategies, the authors are asked to clarify whether the listed recommendations resulted from the research conducted.

 Author response:  Thank you very much for the comment. They should be linked, but we could used unclear sentences.

Author action: Based on the comment from the respected reviewer, the recommendations were enhanced.